# Epigenetic Regulation in Oral Squamous Cell Carcinoma Microenvironment: A Comprehensive Review

**DOI:** 10.3390/cancers15235600

**Published:** 2023-11-27

**Authors:** Hassan Mesgari, Samar Esmaelian, Kamyar Nasiri, Shabnam Ghasemzadeh, Parisa Doroudgar, Zahra Payandeh

**Affiliations:** 1Oral and Maxillofacial Surgery Department, Faculty of Dentistry, Islamic Azad University Tehran Branch, Tehran 1148963537, Iran; kmesgari@gmail.com; 2Faculty of Dentistry, Islamic Azad University Tehran Branch, Tehran 1148963537, Iran; samar.esmaelian@gmail.com; 3Faculty of Dentistry, Islamic Azad University of Medical Sciences, Tehran 1148963537, Iran; kamyar.nasiri.dds@gmail.com; 4Faculty of Dentistry, Qazvin University of Medical Sciences, Qazvin 1461965381, Iran; ghasemzadeh.shab@gmail.com; 5Department of Oral Medicine, School of Dentistry, Tehran University of Medical Sciences, Tehran 1148963537, Iran; 6Department of Molecular Biosciences, Wenner-Gren Institute, Stockholm University, SE 106 91 Stockholm, Sweden

**Keywords:** oral squamous cell carcinoma, epigenetic, DNA methylation, tumor microenvironment, drug resistance

## Abstract

**Simple Summary:**

This comprehensive review focuses on the role of epigenetics in oral squamous cell carcinoma (OSCC), a prevalent type of oral cancer. The review highlights the importance of epigenetic changes, including DNA methylation, histone modifications, and miRNAs, in OSCC development and progression. Aberrant DNA methylation of tumor suppressor genes (TSGs) promotes tumor growth, making gene methylation patterns potential biomarkers for OSCC detection. Histone modifications, such as acetylation, methylation, phosphorylation, and ubiquitination, impact gene expression by modifying chromatin structure. Dysregulated miRNAs also contribute to OSCC progression. Epigenetic-targeted therapies, such as DNMT and HDAC inhibitors, show promise in modifying abnormal gene expression patterns, potentially leading to improved treatment outcomes for OSCC. However, challenges remain in biomarker identification and developing effective combination treatments. Understanding and targeting these epigenetic processes offer potential strategies to overcome drug resistance and improve OSCC treatment efficacy. Overall, the review highlights the potential of understanding and targeting epigenetic processes to overcome challenges and improve the efficacy of OSCC treatment, offering valuable insights for society in the diagnosis and treatment of oral cancer.

**Abstract:**

Oral squamous cell carcinoma (OSCC) is a prevalent and significant type of oral cancer that has far-reaching health implications worldwide. Epigenetics, a field focused on studying heritable changes in gene expression without modifying DNA sequence, plays a pivotal role in OSCC. Epigenetic changes, encompassing DNA methylation, histone modifications, and miRNAs, exert control over gene activity and cellular characteristics. In OSCC, aberrant DNA methylation of tumor suppressor genes (TSG) leads to their inactivation, subsequently facilitating tumor growth. As a result, distinct patterns of gene methylation hold promise as valuable biomarkers for the detection of OSCC. Oral cancer treatment typically involves surgery, radiation therapy, and chemotherapy, but even with these treatments, cancer cells cannot be effectively targeted and destroyed. Researchers are therefore exploring new methods to target and eliminate cancer cells. One promising approach is the use of epigenetic modifiers, such as DNA methyltransferase (DNMT) inhibitors and histone deacetylase (HDAC) inhibitors, which have been shown to modify abnormal epigenetic patterns in OSCC cells, leading to the reactivation of TSGs and the suppression of oncogenes. As a result, epigenetic-targeted therapies have the potential to directly alter gene expression and minimize side effects. Several studies have explored the efficacy of such therapies in the treatment of OSCC. Although studies have investigated the efficacy of epigenetic therapies, challenges in identifying reliable biomarkers and developing effective combination treatments are acknowledged. Of note, epigenetic mechanisms play a significant role in drug resistance in OSCC and other cancers. Aberrant DNA methylation can silence tumor suppressor genes, while alterations in histone modifications and chromatin remodeling affect gene expression related to drug metabolism and cell survival. Thus, understanding and targeting these epigenetic processes offer potential strategies to overcome drug resistance and improve the efficacy of cancer treatments in OSCC. This comprehensive review focuses on the complex interplay between epigenetic alterations and OSCC cells. This will involve a deep dive into the mechanisms underlying epigenetic modifications and their impact on OSCC, including its initiation, progression, and metastasis. Furthermore, this review will present the role of epigenetics in the treatment and diagnosis of OSCC.

## 1. Introduction

Oral squamous cell carcinoma (OSCC) is a form of cancer that affects the cells that line the oral cavity, encompassing the lips, tongue, gums, floor of the mouth, and inner lining of the cheek [1]. It is the predominant form of oral cancer, constituting over 90% of all oral malignancies [2]. In certain regions of the world, it has a fatality rate of approximately 50% [3].

In Western countries, OSCC primarily affects the tongue in 20% to 40% of cases and the floor of the mouth in 15% to 20% of cases, together accounting for approximately 50% of all cases of OSCC [4,5]. OSCC has a higher incidence in men compared to women (male-to-female ratio of 1.5:1), likely due to a higher prevalence of high-risk behaviors among men. The risk of developing OSCC is influenced by the duration of exposure to risk factors, and advancing age introduces additional mutagenic and epigenetic changes. In the United States, the median age of diagnosis for OSCC is 62 years, but there is an increasing incidence of OSCC among individuals under the age of 45, although the reasons for this trend are not well understood [6].

OSCC symptoms include recurrent mouth sores or ulcers that do not heal, red or white patches in the mouth, pain while swallowing, a chronic sore throat, a lump or mass in the mouth or neck, ear ache, and changes in speech or voice [7]. Notably, if any of these symptoms are present, it is important to consult a health care professional for further evaluation.

OSCC is diagnosed through a thorough examination of the oral cavity and imaging tests like X-rays, CT scans, or MRI scans, not to mention that a biopsy is necessary to confirm the diagnosis and stage the cancer [8,9]. Early detection and prompt treatment are crucial for improving OSCC prognosis. Treatment for OSCC depends on tumor stage, location, and overall health. For instance, the stage of a tumor, which indicates its size and extent of spread, is an important factor in determining the treatment approach [10]. Early-stage OSCC (stages I and II) is often treated with surgery or radiation therapy, while advanced-stage OSCC (stages III and IV) may require a combination of surgery, radiation therapy, and chemotherapy [10,11]. However, options may include surgery to remove the tumor and nearby tissues, radiation therapy to destroy cancer cells, chemotherapy to target cancer cells throughout the body, targeted drug therapy, or a combination of these treatments [12]. These methods may lead to side effects such as mucositis, oral candidiasis, loss of gustatory sensation, xerostomia [13], increased risk of infection, salivary gland dysfunction, taste dysfunction, pain [14], and various other specific effects depending on the treatment modality used.

OSCC commonly arises from squamous cells, which are flat, thin cells that form the lining of the oral cavity [15]. While the precise mechanism behind OSCC is not completely comprehended, specific risk factors can elevate the chances of developing this condition. These risk factors include tobacco use (smoking or chewing), heavy alcohol consumption [16], human papillomavirus (HPV) infection [17,18], prolonged exposure to sunlight (lip cancer) [19], a weakened immune system [20], poor oral hygiene, and a history of oral precancerous lesions [21].

Notably, epigenetics, the investigation of heritable modifications in gene expression or cellular traits that do not entail changes to the underlying DNA sequence, has been discovered to have a crucial role not only in OSCC progression [22] but also in the diagnosis and potential treatment thereof [23].

Epigenetic alterations consist of DNA methylation, modifications to histones, regulation facilitated by non-coding RNA, chromatin remodeling, and genomic imprinting [24]. Together, these epigenetic mechanisms dynamically control gene activity and cellular phenotypes, playing essential roles in development, disease processes, and the interplay between genes and the environment [25]. For instance, aberrant DNA methylation of tumor suppressor genes (TSG), such as DAPK, p16, APC, MGMT, TIMP3, CDH1, and so forth has been observed in OSCC [23,26,27,28,29]. As a result, these DNA methylation changes can lead to the silencing of these genes, promoting uncontrolled cell growth and tumor progression [30].

In addition, the examination of DNA methylation patterns in particular genes has been explored as potential indicators for OSCC. For instance, the methylation statuses of genes such as SFRP1 [31], MGMT [32,33], and CDKN2A [34] have been found to be altered in OSCC, and these changes can be detected in patient samples. Thus, analyzing the methylation patterns of these genes can aid in the diagnosis and early detection of OSCC.

Most importantly, epigenetic modifiers such as DNA methyltransferase (DNMT) inhibitors [35] and histone deacetylase (HDAC) inhibitors [36] have been explored as potential therapeutic agents for OSCC. Indeed, these drugs can modify the abnormal epigenetic patterns observed in OSCC cells, leading to reactivation of TSG or silencing of oncogenes. Further, they can be used because they directly affect altering gene expression and have indirect effects on reducing side effects [37,38]. All these events also occur for miRNAs, since they can contribute not only to OSCC pathogenesis but also to OSCC treatment [39,40]. 

As briefly discussed above, epigenetics plays a pivotal role in OSCC, highlighting its complex involvement in cancer progression and treatment. The role of epigenetics in OSCC underscores the complexity of epigenetic regulation and highlights the potential of epigenetic-targeted therapies for managing this challenging disease. In the remainder of this review article, aspects of epigenetics in OSCC, to the best of our knowledge, are reviewed and discussed by delving into the mechanisms underlying epigenetic modifications and their contributions to OSCC initiation, progression, and metastasis, as well as treatment and diagnosis.

## 2. Microenvironment of OSCC

Oral cancer treatment commonly involves surgery, radiotherapy, and chemotherapy. However, developing effective strategies is challenging due to the complex tumor microenvironment (TME), which influences tumor growth and response to treatment [41]. The TME can promote tumor progression, hinder drug delivery, and suppress the immune system’s response. Researchers are exploring ways to modulate the TME and improve treatment outcomes, including through immunotherapy [42].

The TME consists of both non-cellular and cellular components. The non-cellular components are primarily found in the extracellular matrix, which is the network of molecules that surrounds and supports cells. The cellular components include fibroblasts (a type of connective tissue cell) and various immune cells [43]. When the TME becomes abnormal, it can have several negative effects on tumor progression and treatment outcomes. For example, an abnormal TME can promote the invasion of tumor cells into surrounding tissues and hinder the diffusion of therapeutic drugs, making them less effective [44].

Furthermore, the TME can exert immunosuppressive effects. It can inhibit the activity of tumor-specific T cells, known as cytotoxic T lymphocytes or CTLs, that are responsible for recognizing and eliminating cancer cells. At the same time, it can promote the accumulation of regulatory T cells (Tregs), which suppress immune responses. This imbalance leads to a decrease in the secretion of immune-activating cytokines and an increase in the secretion of immunosuppressive cytokines, creating an environment that helps tumor cells evade immune surveillance. The abnormal TME can also hinder the function of dendritic cells, which are important in initiating immune responses by presenting antigens to T cells [45]. 

The occurrence, development, and metastasis of tumors, including OSCC, are influenced by both the external environment surrounding the tumor cells and the internal environment within the cancer cells themselves. The TME provides necessary resources for tumor growth and can also impede tumor growth and metastasis through immune responses and physical barriers. The relationship between tumors and their microenvironment is complex, with tumor cells actively modifying the microenvironment to their advantage, while the microenvironment can also exert antagonistic effects on tumor cells [46]. 

The extracellular matrix (ECM) is a complex network of macromolecules that surrounds cells and provides structural and biochemical support to surrounding cells. It is composed of various types of macromolecules, including collagen, proteoglycans, glycoproteins, and glycolipids, which are secreted by cells and deposited in the extracellular space [47]. Tumor cells can modulate the composition and organization of the ECM to promote their own growth and survival. The ECM can provide a scaffold for tumor cells to grow and invade surrounding tissues, and it can also influence the immune response against cancer cells. Understanding the complex interactions between tumor cells and the ECM is essential for developing effective cancer therapies [48]. Abnormalities in ECM regulation is a prominent feature of the TME [49]. Key components of the ECM include collagen, fibronectin, elastin, and hyaluronic acid. Collagen enhances tissue strength and is degraded by enzymes called matrix metalloproteinases (MMPs) [50]. Fibronectin, a key protein in the ECM, plays a crucial role in various cellular processes such as cell adhesion, migration, proliferation, and vascularization. It interacts with growth factors like IGF, FGF, TGF-β, HGF, and PDGF [51,52]. Elastin provides tissue strength and elasticity [53], while hyaluronic acid regulates vascular permeability, wound healing, and material diffusion [54]. The ECM acts as a catalyst for growth factor activity, serving as a repository for these molecules and promoting their interaction with receptors. ECM degradation releases growth factors and cytokines, including MMP and VEGF [55], which contribute to tumor development. PDGF can accumulate in ECM by binding with collagen, and HBGF-1 binds to type I and type IV collagen [56]. Overall, ECM components and their interactions with growth factors play important roles in tumor development.

Notably, TME is closely connected to tumor angiogenesis, which is the process of forming new blood vessels to provide oxygen and nutrients to the tumor. As a tumor grows beyond 2 mm in size, it faces challenges in obtaining sufficient oxygen and nutrients [57]. In response, tumor cells release angiogenic factors into the TME, triggering the development of new blood vessels. Tumor cells can directly or indirectly contribute to the formation of blood vessels. In an oxygen-deprived environment, a protein called hypoxia-inducible factor (HIF) increases the expression of angiogenic factors such as VEGF [58]. Tumor cells also recruit and transform other cells into tumor-associated stromal cells, which stimulate the production of vascular growth factors. VEGF binds to VEGFR in tumor tissues, activating signaling pathways that alter vascular permeability and promote angiogenesis. Tumor angiogenesis is strongly linked to tumor growth and the spread of cancer cells. Inhibiting or disrupting the tumor angiogenesis microenvironment is a potential approach to treating tumors by interfering with their blood supply [59,60].

In addition, stromal cells, including cancer-associated fibroblasts (CAFs), endothelial cells, adipocytes, and macrophages, play important roles in tumor development. CAFs, derived from fibroblasts and mesenchymal stem cells [61], can differentiate into various cell types and secrete factors that promote tumor cell proliferation, invasion, and angiogenesis [62]. Endothelial cells contribute to the formation of new blood vessels within tumors, supporting their growth and metastasis. Adipocytes secrete adipokines that impact tumor initiation and progression, while macrophages exist in different subtypes, with some promoting inflammation and tumor suppression (M1) [63] and others supporting tissue remodeling and angiogenesis (M2) [64]. The interactions between stromal cells and tumor cells within the TME are complex. Stromal cells have been found to influence tumor growth, invasion, immune responses, and other processes that facilitate tumor progression [65,66,67]. 

Tumor-infiltrating lymphocytes (TILs) are immune cells that enter tumors in response to the body’s immune system fighting against the tumor [68]. TILs consist of T lymphocytes, B lymphocytes, and natural killer (NK) cells. T lymphocytes are critical for the anti-tumor response and are dominant in the TME. They can activate CD8+ T cells and tumor-specific antibodies, inhibit tumor growth, and support the activation of macrophages and the maturation of dendritic cells (DCs) [69]. Tregs, a specific type of CD4+ T cells, are attracted to the tumor area by chemical signals released by tumor cells and macrophages, and they help prevent autoimmune diseases [70,71,72]. NK cells, which originate from lymphoid stem cells in the bone marrow and mature in the bone marrow and thymus, regulate the adaptive immune response by interacting with DC cells and displaying a strong immune response against tumor cells [73]. However, tumor cells have developed ways to resist NK cell attacks, such as releasing TGF-β, reducing the expression of recognizable antigens, and increasing the expression of a protein called MHC I [74,75]. Nevertheless, tumor-infiltrating NK cells can still hinder the spread of tumor cells through the bloodstream [75]. B lymphocytes make up approximately 15–20% of TILs and can differentiate into plasma cells when stimulated by antigens [76]. They produce antibodies and contribute to the activation of the humoral immune response [77]. (Figure 1).

Notice that cell-to-cell communication within the TME is vital for the progression of tumors. One important mechanism of communication is through exosomes, which are small sacs released by cells and surrounded by a lipid bilayer membrane [79]. Exosomes play a significant role in reshaping the TME by facilitating communication between cells and regulating processes like angiogenesis, immune evasion, and distant metastasis. Exosomes can have both stimulating and inhibiting effects on the immune system. They can carry tumor-associated antigens, which can trigger specific immune responses against the tumor [80]. This can have an anti-tumor effect by activating immune cells to target and eliminate cancer cells. Additionally, exosomes can also activate innate immune responses, which are the body’s general defense mechanisms against infections and diseases, which lead to the elimination of cancer cells [81,82]. 

In studies conducted on mice with precancerous lesions of OSCC, it was observed that a single injection of exosomes derived from tumors could actually accelerate tumor progression. This suggests that tumor-derived exosomes can have complex effects on the TME and influence tumor growth and development [83].

## 3. Epigenetic Alterations in OSCC

Epigenetic changes observed in OSCC, such as DNA methylation, histone modifications, and non-coding RNA expression, play a substantial role in the advancement and progression of the disease. These alterations govern critical genes associated with cell DNA repair, cycle regulation, metastasis, and apoptosis. Hence, understanding and targeting these epigenetic alterations hold promise for novel diagnostic and therapeutic strategies in OSCC management, potentially improving patient outcomes.

### 3.1. DNA Methylation

DNA methylation is a cellular mechanism that adds a methyl group to the DNA at cytosine residues within CpG dinucleotides. The DNMT family (DNMT1, DNMT2, DNMT3A, DNMT3B, and DNMT3L) catalyzes this process [84,85], which recognizes specific DNA sequences and produces 5-methylcytosine, which is crucial for gene regulation and its chromatin structure [86] (Figure 2).

Both hypermethylation and hypomethylation play crucial roles in the development of OSCC. Hypomethylation can lead to chromosomal instability and reactivation of silenced genes, including proto-oncogenes, which accelerate the progression of cancer. Additionally, hypomethylation may contribute to oral carcinogenesis through loss of imprinting, resulting in altered gene expression [88]. Although the impact of a hypomethylation-induced loss of imprinting in OSCC remains to be investigated, it has been observed in other tumor types [89]. Likewise, promoter hypermethylation can silence TSG, impair DNA repair mechanisms, and promote immune evasion; not to mention that most hypermethylated genes have been found to prevent OSCC in some cases. These epigenetic alterations are key molecular events in OSCC tumorigenesis and provide potential targets for diagnostic and therapeutic strategies [90] (Figure 3).

#### 3.1.1. Hypomethylated Genes Involving OSCC

DNA hypomethylation pertains to a reduction in the methylation of cytosine residues within DNA molecules. Methylation is a chemical alteration of DNA wherein a methyl group (CH3) is appended to the cytosine base, usually taking place at CpG sites—regions where cytosine is followed by guanine [88]. Accordingly, it plays a crucial role in gene regulation and silencing, which in turn leads to altered gene expression patterns, increased expression of normally silent genes, and potential genomic instability [91]. Hypomethylation contributes to tumor development and progression in OSCC and other cancers by promoting the expression of genes that drive cell growth, invasion, and metastasis [89]. 

Tobacco usage, a prominent risk factor for OSCC, has been correlated with general global hypomethylation and a decrease in DNA methylation levels across the entire genome. This alteration in DNA methylation patterns can disrupt gene expression, potentially activating oncogenes and inactivating TSG, leading to uncontrolled cell growth and OSCC development [92,93]. The mechanisms underlying global hypomethylation due to tobacco exposure are not fully understood, but may involve direct chemical interference or the generation of oxidative stress and inflammation [94]. 

Hypomethylation refers to a decrease in the level of DNA methylation that can have significant effects on gene expression. In cancer, hypomethylation can promote tumor progression by demethylation of previously methylated promoter regions of various oncogenes [95]. This demethylation alters the expression of these oncogenes, potentially leading to their increased activity. Hypomethylation and the resulting altered expression of oncogenes are associated with the progression of malignant tumors. This is because hypomethylation-induced changes can contribute to genomic instability, making it more susceptible to genetic abnormalities and mutations. These genetic instabilities can further drive the development and progression of cancer by disrupting normal cellular functions and promoting uncontrolled cell growth [22]. Cyclins [96], EGFR [97], AIM2 [98,99], CEACAM1 [100], LINE-1 [101], PI3 [102], and PTHLH [103] are the most important hypomethylated genes that have been found to contribute to OSCC [104]. These genes are discussed in the following subheadings. 

##### Cyclins

Cyclin D1, derived from PRAD1 or CCND1 located on chromosome 11q13, acts as a promoter of the cell cycle. In normal cells, cyclin D1 promotes the progression of the cell cycle via the G1 phase. However, excessive cyclin D1 production can lead to a shortened G1 phase, increased cell proliferation, and reduced dependence on growth factors. Elevated levels of cyclin D1 have been detected in various tumor types, including hepatocellular, head and neck, esophageal, lung cancers, and OSCC [105,106,107].

Of note, within the D-type cyclin family, which includes D1, D2, and D3 isoforms, only cyclin D1 is expressed in cases of oral cancer [96]. In cancer, hypomethylation often correlates with the overexpression of oncogenes, genes that promote cell growth, or other genes involved in tumor development and progression. The activation of these genes can occur as a result of hypomethylation, which contributes to enhanced cell proliferation, decreased reliance on growth factors, and other cancer cell characteristics. Hypomethylation of the CCND1 promoter region has been observed in OSCC, leading to increased Cyclin D1 expression [108]. Elevated levels of Cyclin D1 can dysregulate the cell cycle and promote uncontrolled cell growth [109].

Immunostaining findings have demonstrated that the protein accumulates within the nucleus in oral cancer. Additionally, elevated levels of cyclin D1 mRNA have been reported in OSCC, and its expression appears to be specific to different stages of the disease [110]. Indeed, there is a correlation between increased levels of cyclin D1 and the severity of the disease, such as tumor stage, lymph node involvement, and aggressiveness of OSCC [111,112]. This means that the amplification and overexpression of Cyclin D1 in patients has been associated with unfavorable outcomes and reduced survival rates [113,114]. 

Cyclin E is a protein that forms a complex with CDK2 to regulate the cell cycle. It plays a crucial role in promoting the transition from the G1 to S phase, where DNA replication occurs. Aberrant regulation of Cyclin E can result in uncontrolled cell proliferation and is linked to a range of diseases, including cancer [115]. Cyclin E is a protein that is known to be unstable and has a short lifespan within cells. However, in cases of dysplasia and OSCC, there is an observed increase in the levels of cyclin E compared to the normal oral epithelium [116,117,118,119,120]. This upregulation is believed to be the result of gene amplification, specifically at a chromosomal locus called 19q12, which contains the cyclin E gene [117]. This amplification leads to an increased number of copies of the cyclin E gene and, subsequently, higher levels of cyclin E protein. Cyclin E gene amplification has been associated with the severity of OSCC, indicating a potential role for cyclin E in disease progression. 

Furthermore, cyclin E has been linked to other cancer-related changes in OSCC cells. It is positively correlated with centrosome amplification, which refers to an abnormal increase in the number of centrosomes within cells. Centrosome amplification is commonly observed in cancer cells and is associated with genomic instability and tumor progression [116]. Furthermore, there was an inverse relationship observed between the expression of cyclin E and p27, a protein responsible for regulating the cell cycle. Diminished levels of p27 are associated with heightened cell proliferation, and this negative correlation implies that cyclin E may play a role in the disruption of the cell cycle and the promotion of cell growth in OSCC [121]. 

Cyclin A is a protein involved in the regulation of the cell cycle and exists in two primary isoforms, namely A1 and A2 [122]. Cyclin A2 is typically found in non-reproductive cells, whereas cyclin A1 is primarily expressed in reproductive cells. However, cyclin A1 overexpression has been reported in a specific subgroup of OSCC cells and laryngeal squamous cell carcinoma (SCC). This suggests that cyclin A1 plays a distinct role in the development and progression of OSCC [123,124,125,126,127]. 

Cyclin B, a protein with multiple isoforms including cyclin B1, B2, and B3, displays irregular expression patterns in OSCC. Specifically, cyclin B1 is significantly upregulated in tongue carcinomas, a subtype of OSCC, and this increased expression is linked to more aggressive characteristics in OSCC [120,125,126,128]. The presence of cyclin B1 has been established as a reliable marker for assessing the degree of tumor proliferation in OSCC [129] and is considered a useful prognostic indicator for predicting lymph node metastasis in tongue carcinomas [130].

Furthermore, overexpression of cyclin B1 in OSCC is closely linked to the clinical outcomes of patients who have received treatment for oral precancerous conditions. As a result, clinicians can predict the likelihood of disease recurrence or progression and tailor treatment plans accordingly by monitoring cyclin B1 expression [131,132]. Altogether, the aberrant expression of cyclin B1, particularly in tongue carcinomas, provides valuable information about the aggressiveness of the cancer and its potential to spread, aiding diagnosis, prognosis, and treatment decision making in OSCC patients.

DNA hypomethylation can result in enhanced transcriptional activity and increased cyclin expression. In turn, this may disrupt the normal balance of cell cycle control and contribute to uncontrolled cell proliferation. However, it is important to note that the specific impact of DNA hypomethylation on cyclin genes may vary depending on the context and specific regulatory elements involved.

##### Epidermal Growth Factor Receptor (EGFR)

EGFR is a receptor tyrosine kinase that plays a pivotal role in regulating cell proliferation, survival, and differentiation. In the context of OSCC, the hypomethylation of the promoter region of the EGFR gene has been associated with increased expression of the EGFR protein. This upregulation of EGFR has notable implications in OSCC, as it promotes cell proliferation, inhibits apoptosis, and contributes to the progression of tumors [97].

In OSCC, the promoter region of the EGFR gene undergoes hypomethylation. This hypomethylation leads to increased activity of the EGFR gene, resulting in elevated EGFR protein [133,134,135]. When the EGFR gene is hypomethylated in OSCC, it leads to a higher production of EGFR protein, which means that it stimulates the uncontrolled growth and division of cancer cells [136,137].

##### (Absent in Melanoma 2) AIM2

AIM2 encodes a cytosolic protein involved in innate immunity and inflammation [138]. It is a member of the AIM2-like receptor family, which plays a crucial role in detecting cytosolic DNA derived from pathogens or cellular damage [139]. The AIM2 protein consists of several domains, including a pyrin domain, DNA-binding HIN-200 domain, and C-terminal oligomerization domain, which in turn enables AIM2 to function as a DNA sensor within the cell [140].

Upon sensing cytosolic DNA, AIM2 (Absent in melanoma 2) associates with the adaptor molecule ASC (apoptosis-associated speck-like protein containing a CARD) and procaspase-1, resulting in the formation of a multiprotein complex known as the inflammasome. Activation of the inflammasome prompts the cleavage and release of inflammatory cytokines, including interleukin-1 beta (IL-1β), and triggers pyroptosis, a programmed cell death mechanism [141,142]. Dysregulation of AIM2 has been implicated in autoimmune diseases, inflammatory disorders, and certain cancers [143], such as OSCC [99].

In normal cells, AIM2 is regulated by DNA methylation, which helps maintain normal expression levels. However, hypomethylationcan lead to increased AIM2 expression [144]. When the AIM2 gene becomes hypomethylated, its regulatory regions may become more accessible to transcription factors, allowing for increased binding and subsequent transcription of the gene, leading to AIM2 overexpression. Hypomethylation-induced overexpression of AIM2 has been observed in various types of cancer, including OSCC. Studies have shown that hypomethylation of AIM2 is associated with increased AIM2 expression in OSCC tissues compared to normal oral tissues [145,146,147].

One potential mechanism of AIM2 in OSCC progression is activation of the AIM2 inflammasome. AIM2 is a cytosolic sensor protein that forms a complex called the AIM2 inflammasome when cytosolic DNA is detected [138,148]. In OSCC, increased AIM2 expression can lead to sustained activation of the AIM2 inflammasome [149], yielding the production of pro-inflammatory cytokines, such as IL-18 and IL-1β [150,151]. As a result, these cytokines can promote chronic inflammation, creating an environment favorable for tumor growth and progression.

In addition to inflammasome activation, AIM2 overexpression might directly affect OSCC cell proliferation and survival. AIM2 has been found to interact with various signaling molecules and pathways involved in cell growth and survival, including p53, NF-κB, [99,145], and caspase-8 [152,153], via some pathways like JAK-STAT, MAPK, and AKT pathways [149]. Dysregulation of these pathways due to AIM2 overexpression can enhance cell survival, inhibit apoptosis, and promote OSCC cell proliferation, ultimately contributing to tumor growth [154,155]. Furthermore, AIM2 overexpression in OSCC leads to increased angiogenesis and invasion indirectly by promoting the secretion of pro-inflammatory cytokines such as IL-18 and IL-1β [141,142], which can induce angiogenesis [156,157], promoting the recruitment of endothelial cells and vascular network support for tumor growth. This invasive potential allows OSCC cells to infiltrate tissues and potentially metastasize [158].

Another potential consequence of AIM2 overexpression in OSCC is the immune evasion. This means that persistent inflammation induced by AIM2 overexpression attracts immunosuppressive cells, including regulatory T cells (Tregs) [159,160], myeloid-derived suppressor cells (MDSCs) [161], and tumor-associated macrophages (TAMs) [162] at the tumor site. Tregs suppress the activity of effector T cells, which are responsible for recognizing and eliminating cancer cells. MDSCs promote immune suppression by inhibiting the function of various immune cells. TAMs can exhibit both proinflammatory and immunosuppressive properties depending on the context [163,164]. The recruitment and accumulation of these immunosuppressive cells contribute to immune evasion in OSCC. Thus, chronic inflammation triggered by AIM2 overexpression could create an immunosuppressive microenvironment. This can lead to the recruitment of immunosuppressive cells and production of factors that inhibit immune cell function. As a result, OSCC cells can evade immune detection and elimination, facilitating tumor progression [165,166].

##### CEACAM1

CEACAM1 is responsible for encoding a protein known as carcinoembryonic antigen-related cell adhesion molecule 1. This protein plays a role in cell adhesion and various signaling processes [167]. It is primarily found on the surface of cells in epithelial tissues and plays a role in cell–cell recognition and communication. CEACAM1 also participates in signal transduction pathways and regulates cellular processes such as cell differentiation, proliferation, and apoptosis. Dysregulation of CEACAM1 has been associated with various cancers and is being studied for potential therapeutic applications in various diseases [168,169]. Some studies have reported hypomethylation of CEACAM1, which leads to its increased expression [170,171]. 

CEACAM1 overexpression in OSCC can promote tumor growth and invasion through various mechanisms [172,173]. One key pathway involves the activation of signaling cascades such as the PI3K/Akt and MAPK/ERK pathways [174,175]. CEACAM1 interacts with receptor tyrosine kinases and activates downstream signaling molecules, leading to enhanced cell survival, proliferation, and resistance to cell death signals. In the research conducted, it was discovered that among the OSCC-derived cell lines, there were 188 genes that showed decreased expression. Notably, CEACAM1 was identified as the most significant gene in this regard, and its downregulation was associated with tumor necrosis factor (TNF) staging, suggesting its potential contribution to OSCC progression [176]. Additionally, CEACAM1 can modulate the expression and activity of matrix metalloproteinases (MMPs) [177] and enzymes involved in extracellular matrix degradation, which facilitates tumor invasion and metastasis by degrading matrix barriers and enhancing angiogenesis [178].

Furthermore, CEACAM1 interacts with natural killer (NK) cells as an inhibitory receptor for NKG2D-mediated cytolysis [179] and can thus deliver inhibitory signals to dampen their cytotoxic activity. It can engage inhibitory receptors on immune cells, including T-cell and NK cell receptors, leading to the downregulation of immune responses [180]. This inhibitory effect can impair the ability of immune cells to recognize and eliminate OSCC cells, allowing tumor cells to escape immune surveillance [181]. 

##### LINE-1

LINE-1 (Long Interspersed Nuclear Element-1), also known as L1, is a repetitive DNA sequence found in mammals, including humans. It constitutes a significant portion of the human genome and can move within the genome through a copy-and-paste mechanism. LINE-1 elements encode proteins that enable their retrotransposition, a process in which they are transcribed into RNA, converted back into DNA, and inserted into new genomic locations [182]. LINE-1 elements have played a role in genome evolution and can influence gene expression, genome stability, and genetic diversity. However, their insertion can also lead to gene disruptions and contribute to diseases, including cancer [183]. Studying LINE-1 elements contributes to our understanding of genome biology and disease development [184].

LINE-1 hypomethylation refers to a specific alteration in the DNA methylation pattern of LINE-1 elements. In normal cells, including healthy somatic cells, LINE-1 elements are heavily methylated, meaning that the cytosine residues within their CpG sites are methylated, thereby helping maintain the stability and integrity of the genome by suppressing the activity of transposable elements like LINE-1 [185]. LINE-1 hypomethylation has been observed in various types of cancer and is considered a common epigenetic alteration associated with tumorigenesis [186,187,188,189]. It can result in increased LINE-1 activity and retrotransposition, as hypomethylated LINE-1 elements are more prone to transcription and subsequent retrotransposition events of LINE-1, which may contribute to tumor development and progression [190,191]. 

The precise mechanisms by which LINE-1 hypomethylation influences cancer development are still under investigation. Notably, L1 retrotransposons encode two proteins known as ORF1p and ORF2p, and the expression of both proteins is crucial for the process of retrotransposition [192]. OSCC was the focus of a study investigating the activity of L1 retrotransposons, genetic elements that can move within the genome. The study analyzed L1 ORF1p expression in OSCC samples from 15 patients and found that approximately 60% of the cancer samples exhibited ORF1p expression, with some showing aberrant p53 expression. The researchers also observed trends of hypomethylation in the L1 promoter region in cancer tissues compared to normal tissues. These findings suggest that the expression of L1ORF1p may contribute to the onset and progression of OSCC [193], though further research is needed to validate and expand upon these initial results. Likewise, in another study, a cohort of 77 patients with advanced oropharyngeal squamous cell carcinoma (OPSCC) was retrospectively reviewed to investigate the methylation of LINE-1 repetitive sequences in their tumor tissues. The researchers aimed to determine if LINE-1 methylation could predict early relapse after treatment. The study revealed that patients who experienced relapse within a two-year timeframe had notably lower levels of LINE-1 methylation compared to those who did not relapse. This correlation was observed in both HPV16-negative and HPV16-positive patients with OPSCC, although statistical significance was only achieved in the HPV16-negative subgroup. Additionally, lower LINE-1 methylation was observed in relapsed cases among current smokers with OPSCC. Through logistic regression analysis, it was determined that patients with reduced LINE-1 methylation had a 3.5 times higher risk of early relapse. These findings were further supported by an independent cohort of 33 OPSCC patients. In summary, the study provides evidence suggesting that decreased LINE-1 methylation is associated with an elevated risk of early relapse in advanced OPSCC cases [194].

Of note, manifold studies have suggested that the p53 protein may regulate the activity of LINE-1 retrotransposons in tumor cells, which enable it to move within the genome and can contribute to tumor development if not properly controlled. The studies propose that p53 protein may silence LINE-1 by influencing the deposition of epigenetic marks, such as DNA methylation or histone modifications, within the LINE-1 promoter region. This regulation of LINE-1 activity by p53 could impact the expression and mobility of these retrotransposons, thus potentially leading to genomic instability and promoting tumorigenesis [195,196,197,198]. The study revealed that OSCC tissues exhibited lower levels of LINE-1 methylation compared to cells obtained from oral rinses of healthy individuals. Interestingly, cells collected from the oral rinses of OSCC patients also displayed a similar degree of LINE-1 hypomethylation as observed in the OSCC tissues. The extent of hypomethylation did not differ based on factors such as cancer stages, locations, histological grades, or the history of betel chewing, smoking, and alcohol consumption. Consequently, it can be concluded that OSCCs demonstrate a global hypomethylation pattern, which can be non-invasively detected through oral rinses using the COBRALINE-1 PCR technique [199]. The same results were also reported by another study; researchers found that low levels of LINE1 global DNA methylation can be associated with worse oral cancer-free survival among patients with OSCC [88]. 

On the whole, studies have shown that in OSCC, there is a reduction in methylation levels specifically within the LINE-1 repetitive elements compared to healthy oral tissues. This LINE-1 hypomethylation is observed regardless of tumor stage, site, grading, or risk factors [199,200]. Consequently, it has the potential to serve as a biomarker for the noninvasive detection of early-stage OSCC. Thus, healthcare professionals may be able to identify individuals at higher risk of developing OSCC by analyzing the methylation status of LINE-1 in easily accessible samples such as oral rinses [199].

##### PI3

The PI3 kinase gene encodes the phosphoinositide 3-kinase (PI3K) enzyme, which is involved in cell signaling pathways that regulate cell growth, survival, and metabolism [201]. Mutations in the PI3K gene, particularly the PIK3CA gene, are associated with various cancers [202] such as OSCC [203] because activation of the PI3K pathway leads to the generation of signaling molecules that promote cell growth and survival. 

The precise role of the hypomethylated PI3 gene in OSCC is not fully understood. Nevertheless, it is known that the PI3K pathway contributes to the development and progression of cancer, including OSCC. For example, a study demonstrated that oridonin, a bioactive compound derived from Rabdosia rubescens with anticancer properties in different types of cancer, inhibited cell proliferation, clonal formation, and induced G2/M cell cycle arrest and apoptosis in OSCC patients in a dose-dependent manner. These effects were achieved by influencing cell cycle proteins and suppressing the PI3K/Akt signaling pathway [204]. 

Accordingly, studies have suggested that two compounds, thymoquinone and licochalcone A, have demonstrated the ability to inhibit the progression of OSCC cells through their effects on the PI3K/AKT pathway. Thymoquinone has been found to suppress the invasion, proliferation, and migration of OSCC cells. Additionally, it induces apoptosis, or programmed cell death, in these cells by inhibiting the PI3K/AKT pathway [205], a signaling pathway involved in cell growth and survival. Similarly, licochalcone A has been shown to suppress the migration, invasion, and proliferation of OSCC cells; these actions are also mediated through modulation of the PI3K/AKT pathway [206].

While the exact relationship between PI3 kinase and hypomethylation in OSCC is currently unclear, there is evidence to suggest that dysregulation of the PI3K pathway can affect other epigenetic modifications and gene expression patterns. It is possible that alterations in the PI3K pathway may indirectly influence DNA methylation patterns, including hypomethylation, through various downstream signaling mechanisms [207,208]. Further research is needed to fully understand the interplay between the PI3K pathway, DNA methylation, and OSCC development, as well as the specific mechanisms underlying hypomethylation in this context.

##### PTHLH

The PTHLH gene encodes a protein called parathyroid hormone-like hormone (PTHrP) [209]. PTHrP is involved in skeletal development, calcium regulation, and various biological processes. It helps in bone growth and remodeling [210], regulates calcium levels in the body, and has roles in embryonic development, tooth eruption, and lactation [211]. Dysregulation of the PTHLH gene can lead to disorders such as hypercalcemia and brachydactyly type E [212,213].

As per a study that investigated the PTHLH gene in OSCC, PTHLH mRNA and protein levels were significantly higher in OSCC cell lines and tissues compared to normal cells. Using siRNA, the downregulation of PTHLH/PTHrP reduced cell proliferation and inhibited colony formation in OSCC cells. Also, changes in cell cycle-related proteins were also observed. Thus, the study suggested that PTHLH/PTHrP can contribute to OSCC development by affecting cell proliferation and the cell cycle, and PTHrP levels may be useful for prognostic evaluation in OSCC [214]. For instance, certain inflammatory cytokines, such as IL-8, IL-6, and TNF-α, have been identified as potential biomarkers for diagnosing oral cancer [215]. Additionally, it states that the PTHLH gene, which codes for the PTHrP, is upregulated in various tumors, including OSCC [216,217]. IL-8, IL-6, and TNF-α are proteins involved in the immune response and inflammation. Studies have suggested that increased levels of these cytokines in saliva may indicate the presence of OSCC, potentially making them useful diagnostic biomarkers [215]. Overall, the passage highlights the potential diagnostic value of IL-8, IL-6, and TNF-α as biomarkers for oral cancer and the upregulation of PTHLH/PTHrP in various tumors, including OSCC (Figure 3).

##### Survivin/BIRC5

Survivin, also known as BIRC5, is a protein that promotes tumor cell proliferation and inhibits apoptosis. It plays a role in enhancing cell division and preventing cell death, allowing cancer cells to survive and multiply [218]. Survivin is frequently overexpressed in various cancers, including OSCC, and its upregulation is associated with aggressive tumor behavior and poor patient outcomes [219]. 

The findings of one study suggested that hypermethylation of the Survivin gene is not observed in OSCC tissues [220]. In normal tissues, this gene is typically methylated, leading to its inactivation. However, in OSCC, the Survivin gene is frequently upregulated due to a hypomethylation. This hypomethylation can induce overexpression of Survivin, which per se promotes cell proliferation and hinders cell death. In a particular study, also, higher levels of Survivin expression were linked to a more aggressive and invasive tumor phenotype in OSCC [221]. Furthermore, in an experimental model using hamster buccal pouch mucosa to study oral carcinogenesis, the findings were that animals treated with mineral oil had Survivin alleles that were methylated, indicating normal gene regulation. Nonetheless, animals treated with the carcinogen 7,12-dimethylbenz[a]anthracene (DMBA) developed buccal pouch carcinomas and showed a hypomethylated Survivin allele [222,223]. These findings suggest that the hypomethylation of the Survivin gene plays a significant role in the development of OSCC. 

Notably, one study [224] specifically focused on an enzyme called nicotinamide N-methyltransferase (NNMT). Previous research has shown that NNMT is upregulated in OSCC and that reducing its expression inhibits cell growth in vitro and tumorigenicity in vivo [225]. But in this study [224], the researchers transfected HSC-2 cells with the NNMT expression vector and evaluated the effect of enzyme upregulation on cell proliferation using the MTT assay. They also investigated the molecular role of NNMT in apoptosis and cell proliferation by examining the expression of β-catenin, survivin, and Ki-67 using real-time PCR. Additionally, immunohistochemistry was performed on 20 OSCC tissue samples to analyze the expression levels of NNMT and the survivin ΔEx3 isoform. The results of the study showed that upregulating NNMT significantly increased cell growth in vitro. Furthermore, there was a positive correlation between NNMT expression levels and survivin ΔEx3 isoform expression levels both in the HSC-2 cells and in the OSCC tissue samples. Overall, these findings suggest that NNMT may play a role in the proliferation and tumorigenic capacity of OSCC cells, indicating that it could potentially be targeted for the treatment of oral cancer. This has clinical relevance as targeting NNMT could potentially improve the survival of OSCC patients by affecting cell growth and anti-apoptotic mechanisms [224].

#### 3.1.2. Aberrant Hypermethylated Genes Involving OSCC

Aberrant hypermethylation refers to an abnormal increase in DNA methylation levels in specific regions of the genome, particularly in CpG islands [226]. It is a common epigenetic alteration observed in various cancers, including OSCC [227,228,229]. Aberrant hypermethylation can result in the silencing of TSG or the activation of oncogenes, disrupting normal gene regulation and promoting tumor development and progression [230]. 

##### CDKN2A

In OSCC, aberrant hypermethylation of the CDKN2A gene, which encodes the p16INK4A protein, is a frequently observed event. Methylation-mediated silencing of CDKN2A/p16 disrupts the normal regulation of the cell cycle [231,232]. The p16INK4A protein acts as a tumor suppressor by inhibiting the activity of cyclin-dependent kinases (CDKs), which are involved in promoting cell cycle progression. When CDKN2A/p16 is hypermethylated and silenced, it leads to uncontrolled cell proliferation and an increased risk of tumorigenesis in the oral cavity [233,234,235]. 

##### MGMT and DAPK1

In addition to the aforementioned, abnormal hypermethylation of the MGMT gene in OSCC has shown to result in the silencing of the DNA repair enzyme known as O^6^-methylguanine DNA methyltransferase, which per se leads to impairing the capacity of cancer cells to fix DNA damage. As a result, this event leads to the accumulation of DNA damage, facilitating further genetic alterations and contributing to the progression of OSCC [236]. Similarly, the aberrant hypermethylation of the DAPK1 gene, involved in programmed cell death and tumor suppression, has been found in OSCC [237], which per se leads to allowing cancer cells to evade apoptosis, enabling their survival and proliferation, thereby promoting tumor development [238,239].

##### TIMP3

Of note, in OSCC, the TIMP3 gene, responsible for regulating extracellular matrix remodeling and cell migration, undergoes aberrant hypermethylation [240,241], which culminates in the silencing of TIMP3 and disrupts the balance between MMPs and their inhibitors [242]. MMPs are a group of enzymes that break down components of the extracellular matrix. Their function is to facilitate the invasion of cancer cells into neighboring tissues and enhance the process of metastasis. The inhibition of TIMP3 through hypermethylation enhances MMP activity, thereby promoting tumor invasion and metastasis in OSCC [243].

##### TFPI2, SOX17, and GATA4

In accordance with a study that focused on understanding a process called transcriptional silencing, which refers to the inactivation of certain genes in a cell, researchers specifically investigated how the promoter regions of 18 candidate TSGs are modified through a process called hypermethylation, leading to their silencing. The TSGs that were studied included MGMT, HIC1, sFRP2, sFRP4, sFRP5, Timp3, p14, p15, p16, TFPI2, sFRP1, SOX17, E-cad, GATA4, GATA5, FBN2, and TCERG1L. The researchers examined 10 different OSCC cell lines and a small set of primary tumor samples from OSCC patients (33 samples in total). The results of the study revealed that most of the genes they investigated were silenced through promoter hypermethylation in the OSCC cell lines. This means that the promoter regions of these genes had excessive methylation, which resulted in the genes being turned off or not expressed. Among the different genes analyzed, TFPI2, SOX17, and GATA4 were frequently found to have hypermethylation not only in OSCC cell lines but also in samples from oral cancer patients. This suggests that these three genes may have a specific role as tumor suppressors in OSCC, meaning they potentially play a critical function in preventing the development or progression of oral squamous cell carcinoma; notably, aberrant hypermethylation thereof can develop OSCC [228].

TFPI2 is a serine proteinase inhibitor that can block the activity of various enzymes involved in biological processes. It has been identified as a TSG in colorectal cancer and other types of cancer [244]. As a TSG, TFPI2 plays a crucial role in inhibiting the formation and progression of tumors. Indeed, its ability to inhibit serine proteases is believed to contribute to its tumor-suppressive effects [245]. Significantly, although the methylation of TFPI2 in OSCC has been detected through a comprehensive methylation array analysis, the connection between TFPI2 promoter hypermethylation and the consequent suppression of gene transcription in OSCC tumor samples has not been confirmed through validation [246]. This means that it is unclear whether TFPI2 promoter hypermethylation leads to the inactivation of TFPI2 gene expression in OSCC. Nonetheless, in one study, the researchers successfully validated that TFPI2 is frequently methylated in a panel of primary tumor samples from OSCC. This methylation pattern is specific to OSCC cells and is not commonly observed in normal cells. Furthermore, they have demonstrated that TFPI2 gene expression is regulated by promoter hypermethylation in OSCC cells [228]. 

SOX17 is a gene responsible for encoding a transcription factor known as HMG box transcription factor. It plays critical roles in diverse biological processes, such as oligodendrocyte development, vascular development, endoderm formation, and embryonic hematopoiesis [247,248,249,250]. The promoter region of SOX17 is often found to be hypermethylated in several types of cancers such as cholangiocarcinoma, lung cancer [251], gastric cancer [252], liver cancer [253], and breast cancer [254]. The hypermethylation of the SOX17 promoter is associated with abnormal activation of the Wnt signaling pathway [255], which is known to contribute to tumorigenesis in multiple cancer types. This abnormal methylation of SOX17 is particularly prevalent in non-small-cell lung cancers (50% of cases) and esophageal squamous cancers (nearly 90% of cases), further supporting its involvement in cancer development [256]. The Wnt signaling pathway is implicated in OSCC, and its abnormal activation, often through mutations or dysregulation of its components, such as β-catenin, is observed in OSCC. This dysregulation leads to uncontrolled cell growth, invasion, and metastasis in OSCC [257]. Additionally, epigenetic modifications, like hypermethylation of the SOX17 gene promoter, can further contribute to the abnormal activation of the Wnt pathway in OSCC [258]. 

The aberrant hypermethylation of the GATA4 gene has been implicated in the emergence of OSCC. GATA4 is a member of the zinc-finger transcription factor family called GATA, which recognizes the GATA motif in gene promoters and plays a crucial role in the development and differentiation of the gastrointestinal tract. In OSCC, GATA4 functions as a TSG that is extensively hypermethylated, leading to the suppression of gene transcription [228]. However, the precise mechanisms and downstream effects of GATA4 hypermethylation in OSCC are still an active area of research.

#### 3.1.3. Hypermethylated Genes Involving OSCC

Hypermethylation is a common occurrence in various types of cancer, including OSCC [259]. Several genetic and environmental factors play a role in OSCC development, of which, here, hypermethylation in the promoter regions of various genes will be discussed.

Methylation of promoter regions of genes such as MGMT, mutL homolog 1 (MLH1), and p15^INK4B^ play significant roles in the development and progression of OSCC. MGMT is crucial for DNA repair [260], and increased levels prevent OSCC, while decreased levels due to methylation make cells vulnerable to carcinogens and mutations [95,261]. MLH1 hypermethylation impairs DNA repair and triggering OSCC by causing changes in genes regulating cell growth and division [95,262]. Hypermethylation of p15^INK4B^ reduces its expression levels leading to abnormal cell growth and division, rendering cells less sensitive to external stimuli, which may allow for OSCC progression [95].

In addition to the genes mentioned above, more than 40 TSGs silenced by hypermethylation on CpG islands and related to OSCC have been described in the literature. The most important of these prevent oral cells from promoting OSCC through hypermethylation, and include E-cadherin [263], which plays a role in cell–cell adhesion and is hypermethylated in OSCC, leading to a loss of cell–cell interactions and cell differentiation. Similarly, the hypermethylation of genes like phosphatase and tensin homolog (PTEN) [264], adenomatous polyposis coli (APC) [265], p14^ARF^ [266], and p16^INK4A^ [267] have been linked to decreased survival rates and poor prognosis in OSCC [104]. 

Nonetheless, hypermethylation is capable of suppressing genes that take part in the advancement and spread of OSCC. An instance is DNMT, whose levels are related to OSCC progression, development, adverse prognosis, and greater chances of metastasis [268]. There is a significant increase in DNMT3a immunoreactivity in OSCC tissues in contrast to normal tissues [269]. Even though some reviews propose typical DNMT1 expression in OSCC, the majority of the studies have proved that the rise of OSCC is linked with DNMT overexpression [92]. Usually, DNMT1 regulates the prognosis of OSCC by reducing their holistic survival [270]. Table 1 shows hypermethylated genes involving OSCC, and in the ensuing subheadings, most important hypermethylated genes involving OSCC will be discussed. 

##### CDKN2A

CDKN2A, also known as cyclin-dependent kinase inhibitor 2A, is a gene located on chromosome 9p21. It encodes two important proteins: p16^INK4a^ and p14^ARF^ [289]. 

The p14^ARF^ gene is a tumor-suppressor gene involved in regulating cell proliferation and division [290,291], as well as tumor-induced angiogenesis [292,293]. When hypermethylation occurs in the p14^ARF^ gene, it leads to the inactivation of its tumor-suppressor functions, including the loss of p53 function and the deactivation of p21-mediated cell proliferation control [294]. This hypermethylation event is associated with advanced stages of carcinogenesis and is linked to increased tumor size, higher tumor stage, and the presence of nodal metastasis [295]. 

In studies conducted on OSCC tumors, it was found that a significant proportion of these tumors, ranging from 14% to 44%, exhibited hypermethylation of the p14^ARF^ promoter [295,296]. Furthermore, a specific study focused on betel quid-related OSCC investigated the role of p14^ARF^ hypermethylation in OPLs. The findings revealed that p14^ARF^ hypermethylation was frequently observed in these early-stage lesions associated with betel quid consumption, whereby the detection of p14^ARF^ hypermethylation can serve as a prognostic marker for the early detection of betel quid-induced OSCC [297]. 

The p16^INK4a^ protein is a tumor suppressor that plays a critical role in regulating the cell cycle. Its primary function is to inhibit the activity of CDK4 and CDK6 [298]. These kinases, when active, initiate the phosphorylation of proteins involved in cell cycle progression. Since p16^INK4a^ inhibits CDK4 and CDK6, it can prevent the phosphorylation of proteins such as retinoblastoma protein (pRb) [299]. Notably, this prevents the release of transcription factors that are necessary for the progression of the cell cycle from G1 phase to S phase. As a result, the cell cycle is arrested in the G1 phase, allowing time for DNA repair, monitoring for cell damage, and cell growth regulation [300]. 

Methylation of the CDKN2A gene is known to affect its expression, resulting in reduced or absent production of the p16^INK4a^ protein. This reduction in p16^INK4a^ levels can disrupt the normal regulation of the cell cycle, potentially leading to uncontrolled cell growth and an increased risk of developing cancer. The methylation rate of CDKN2A has been extensively studied in various types of cancer [291,301,302]. The reported incidence of p16^INK4a^ hypermethylation in OSCC can range from 23% to 76%. This means that the percentage of OSCC cases where the p16^INK4a^ gene is found to be hypermethylated can vary widely in different studies. Hypermethylation of p16^INK4a^ can lead to the inactivation of the gene, potentially contributing to the development or progression of OSCC [32,229,282,303,304,305,306]. Of note, some studies have examined the hypermethylation of p16^INK4a^ in oral mucosa with varying degrees of dysplasia, which are known as pre-neoplastic lesions or oral intraepithelial neoplasia (OIN) [297,307]. Dysplasia refers to abnormal cell growth that can occur before cancer develops [308]. 

Notably, in investigations of OSCC, the presence of abnormal methylation in the p16^INK4a^ TSG was detected in a range of 12% to 88% of the OSCC cases tested [261,295,296,306,309,310,311]. Other studies also found that hypermethylation of p16^INK4a^ occurs more frequently in OIN than in normal mucosa, but less frequently than in OSCC. This suggests that the hypermethylation of p16^INK4a^ is a progressive event in the development of oral cancer, starting from pre-neoplastic lesions to OSCC [297,307].

##### E-Cadherin and N-Cadherin

E-Cadherin and N-Cadherin are cellular adhesion molecules that participate in the adhesion between cells and the organization of tissues [312]. E-Cadherin is found in epithelial tissues and helps hold adjacent cells together, while N-Cadherin is primarily expressed in neural tissues and plays a role in neural development and synaptic connections. Both cadherins mediate calcium-dependent binding between cells, and dysregulation of E-Cadherin and N-Cadherin functions has been reported to contribute to conditions like cancer metastasis [313,314].

In OSCC, hypermethylation of the E-Cadherin (CDH1) gene promoter region has been frequently reported [288,315,316,317]. Promoter hypermethylation of CDH1 can lead to a decrease in or loss of E-Cadherin expression in OSCC cells. E-Cadherin downregulation is associated with reduced cell–cell adhesion, disruption of tissue integrity, and increased invasiveness and metastasis in OSCC. A loss of E-Cadherin function is considered a hallmark of the epithelial–mesenchymal transition (EMT), a process involved in tumor progression and metastasis [315,318]. The reported incidence of CDH1 epigenetic-related modifications in various studies can vary significantly, with values ranging from 7% to 46%. This inconsistency in results suggests a need for standardization of the immunohistochemistry (IHC)-based methods used to assess CDH1 modifications [319]. This indicates that more research is needed to better understand and establish consistent findings regarding the epigenetic changes in CDH1. 

In a recent investigation, researchers explored the expression of N-cadherin, a calcium-dependent adhesion protein, in 94 cases of OSCC. The results demonstrated that OSCC tissue exhibited notably elevated levels of N-cadherin expression, primarily observed in the cytoplasm of cancer cells, in comparison to normal tissue. Furthermore, tumors displaying high levels of N-cadherin were linked to a more aggressive behavior, including increased invasiveness and greater potential for metastasis. These findings suggest that the expression of N-cadherin may serve as a potential indicator for predicting the biological behavior of OSCC, indicating a heightened risk of aggressive tumor growth and metastasis [320]. However, further research is required to fully understand the mechanisms and establish N-cadherin as a reliable predictive marker for OSCC.

Taken together, in OSCC, hypermethylation of the CDH1 is often associated with its downregulation, leading to reduced cell adhesion and increased invasiveness. Conversely, CDH2 is frequently upregulated in OSCC, promoting a more aggressive behavior of cancer cells. These molecular changes contribute to the progression and metastasis of OSCC. This indicates that more research is needed to better understand and establish consistent findings regarding the epigenetic changes in CDH1 and CDH2.

##### PTEN

PTEN (phosphatase and tensin homolog deleted on chromosome 10) is a tumor-suppressor gene found on chromosome 10q23.3 [321]. This gene has been shown to regulate various cellular processes, including survival, differentiation, proliferation, apoptosis, and invasion. The expression of PTEN acts as a control mechanism to prevent uncontrolled cell growth and tumor formation. Loss of PTEN expression, often due to genetic mutations or deletions, disrupts its tumor-suppressor function and can contribute to the development and progression of various types of cancers [322]. 

The loss of PTEN expression leads to a lack of control over certain signaling pathways like Ras/phosphoinositide 3-kinase (PI3K)/AKT involved in apoptosis and migration [323,324,325] (Figure 4). PTEN normally acts as a negative regulator of this pathway, keeping it in check; however, when PTEN expression is lost, the Ras/PI3K/AKT pathway becomes dysregulated, thus promoting tumor cell survival and allowing cancer cells to evade apoptosis and continue proliferating [325,326,327]. (Figure 5).

Kurasawa et al. conducted a study where they analyzed the immunohistochemical expression of PTEN in 113 cases of OSCC and nine OSCC cell lines. The results showed a notable disparity in PTEN expression between tumor samples and normal tissues. Although no mutations were identified, lower levels of PTEN mRNA were observed in four out of six OSCC cell lines. These findings provide support for the theory that PTEN plays a critical role in the development of OSCC, and its downregulation may be linked to hypermethylation [329]. Further, multiple studies have investigated the role of PTEN in OSCC and have found a correlation between abnormal PTEN expression and the occurrence, development, and invasion of OSCC [330].

A study on oral tongue carcinomas revealed that a significant number of tumors exhibited decreased E-cadherin expression, which was linked to hypermethylation of the E-cadherin promoter. This decrease in expression indicated a poor prognosis, and the pattern was consistently observed across primary, recurrent, and metastatic lymph nodes. [331]. Moreover, the same finding was reported in another study; the study found that reduced expression of E-cadherin in invasive and metastatic areas of OSCCs is linked to methylation of its promoter region. Additionally, the study revealed that decreased expression of membranous beta-catenin in these areas is due to protein degradation [332]. Of note, in a review by Diez-Perez et al., they present data from a comparison study between oral cancer tissue and normal mucosa. The study focused on examining the gene expression of a specific gene and found a significant reduction of 77.8% in its expression in oral cancer tissue compared to normal mucosa. This reduction in gene expression was attributed to the methylation of the gene’s promoter region [333]. Indeed, the percentage of CDH1 methylation in OSCC tissues, in accordance with several studies, might range from 17% to 85% [261,316,331,334].

Nonetheless, the study of Squarize et al. suggests that PTEN may not significantly contribute to the development or progression of aggressive forms of OSCC, despite its role as a tumor suppressor gene. This suggests that PTEN expression may not be prevalent in aggressive cases. [335]. Another study aimed to investigate the presence of homozygous deletion of the PTEN gene in OSCC. The researchers analyzed OSCC cell lines and tumor samples using PCR and DNA sequencing methods. Contrary to previous findings in other tumor types, none of the samples exhibited homozygous deletion of PTEN. These results suggest that the homozygous deletion of PTEN is unlikely to be a common occurrence in OSCC [336]. Likewise, the same results were also reported in other studies, not supporting the link between PTEN and OSCC [286,330]. 

These findings provide evidence that promoter methylation of the PTEN gene may not serve as a significant factor in regulating gene expression in OSCC. That is why it is important to note that further studies are necessary to obtain a conclusive understanding of the role of PTEN promoter methylation in OSCC by considering larger sample sizes and diverse patient populations. 

##### P53

The TP53 gene is located on chromosome 17p13.1 and it plays a crucial role in regulating cell processes as a tumor suppressor gene [337]. It encodes a protein known as p53, which is involved in a wide range of fundamental cellular activities. These activities include controlling cell-cycle progression, facilitating cellular differentiation, participating in DNA repair mechanisms, and promoting apoptosis [338]. Accordingly, dysregulation or mutations in the TP53 gene can result in the loss of p53 function, which is associated with increased susceptibility to the development of various types of cancers [339]. P53 mutations occur in between 25% and 69% of cases in the majority of human malignancies, including OSCC [340,341,342,343].

In certain cases, the loss of function of the p53 protein occurs not as a result of genetic mutations but due to epigenetic events. An instance of this is the epigenetic suppression of the non-mutated p53 protein by the E6 protein of high-risk HPVs, specifically HPV16 and HPV18. This occurrence has been observed in both OSCC and certain laryngeal cancers. The E6 protein, produced by high-risk HPVs, can interact with cellular proteins responsible for epigenetic regulation, resulting in the silencing or inactivation of the p53 protein. This epigenetic mechanism disrupts the normal functioning of p53 and contributes to the initiation and advancement of these HPV-related cancers [344].

##### DAPK1

The DAPK1 (Death Associated Protein Kinase 1) gene is located on chromosome 9q34.1 [345]. It is responsible for encoding a protein known as DAPK1, which is a serine/threonine kinase that has been reported to induce apoptosis, a process of programmed cell death. DAPK1 is regulated by calcium and calmodulin, and it acts as a pro-apoptotic factor. It is involved in activating the p53-dependent apoptotic checkpoint, which is a mechanism that helps regulate cell death in response to cellular stress or damage [346]. Promoter hypermethylation of the DAPK gene in OSCC, ranging from 18% to 27%, has been observed in various studies. The reported frequency indicates that a significant proportion of cases show this hypermethylation event [347,348,349]. Of note, a meta-analysis of DAPK methylation status was conducted, which included data from five studies involving a total of 330 cases of OSCC. The analysis found that the overall estimated pooled prevalence of DAPK methylation was 39.7%, with a confidence interval (CI) ranging from 15.0% to 64.3%. It is important to note that there was significant heterogeneity in the results among the studies included in the meta-analysis [350]. In another study, the prevalence of DAPK promoter hypermethylation in primary OSCC tissues accounted for 45.3%. Indeed, DAPK was found to mediate apoptosis by affecting interferon-γ, and its promoter hypermethylation can disrupt this apoptotic pathway, whereby DAPK promoter hypermethylation may contribute to potential metastasis in OSCC primary tumors [351].

##### MGMT

The MGMT (O^6^-methylguanine-DNA methyltransferase) gene is situated on chromosome 10q26. It is responsible for encoding the MGMT protein, which functions as a DNA repair enzyme [352]. MGMT plays a crucial role in removing adducts, or chemical modifications, on DNA caused by alkylating agents. This DNA repair activity is essential for maintaining genome integrity. Interestingly, the activity of MGMT can confer resistance to apoptosis induced by certain treatments. When the MGMT gene is silenced or inactivated, alkylated guanine accumulates in the DNA, which subsequently promotes apoptosis, a programmed cell death process [282]. Silencing MGMT can therefore sensitize cells to apoptosis and potentially enhance the effectiveness of alkylating agent-based treatments [306,353].

Silencing of the O^6^-methylguanine-DNA methyltransferase through promoter hypermethylation is considered an early event in the development of OSCC. Promoter hypermethylation involves the MGMT gene, leading to the inactivation or reduced expression of MGMT [306]. In OSCC, it has been observed that approximately 75% of OSCC tissues exhibit MGMT gene silencing through hypermethylation of its promoter region [354].

While CpG methylation is a well-known mechanism for silencing MGMT expression, it is important to note that other factors may also contribute to the regulation of MGMT gene expression [355]. For example, betel quid chewing, a known risk factor for OSCC, has been associated with a lack of MGMT expression. This suggests that additional factors beyond methylation, such as environmental exposures, may impact MGMT gene expression in OSCC [354]. Of note, the hypermethylation of the MGMT promoter has been found to be associated with poorer survival outcomes in patients with OSCC [309]. For instance, in a study, it was found that the MGMT gene was hypermethylated in tumors that had metastasized in OSCC cases [351]. MGMT is a protein involved in DNA repair, specifically removing adducts from a specific DNA base called O^6^-alkylguanine. If these adducts are not removed, they can cause errors during DNA replication, leading to G-to-A mutations. Therefore, when MGMT function is lost, there is an increased rate of mutations. The hypermethylation of the MGMT gene is strongly linked to certain changes in the K-ras gene at specific locations (codons 12 and 13), as well as lymph node invasion, tumor stage, and disease-free survival [356]. In this study, a significant connection between MGMT methylation and the occurrence of metastases was reported [351]. 

Of note, multiple independent studies have examined OSCC tissues and reported varying frequencies of MGMT methylation, ranging from 7% to 74% [33,261,295,306,309,316,348,357]. These findings suggest that the methylation status of the MGMT gene in OSCC tumors could serve as a valuable diagnostic tool and potentially as a predictor of patient survival. 

##### RARβ2

The RARβ2 (retinoic acid receptor β2) gene is a TSG that plays a role in regulating cell proliferation in various types of cancer and preneoplastic lesions. It belongs to the RARB family of genes and is located on chromosome 3p24 [358]. One common mechanism of inactivation of the RARβ2 gene in cancer is through a process called methylation, which leads to RARβ2 gene silencing or its reduced expression; as a result, it has been observed in several types of cancer [359,360,361].

Likewise, the role of RARβ (retinoic acid receptor β) promoter methylation has generated significant interest in the context of OSCC. For instance, in a pyrosequencing study conducted on OSCC tissue samples, a high frequency of RARβ promoter methylation was observed, with 73% of the samples showing methylation. Interestingly, aberrant methylation of RARβ was also noted in adjacent normal tissues, albeit at a slightly lower frequency of 62% [334]. Of note, this finding indicates that RARβ promoter methylation may not be specific to cancer cells and can also occur in the surrounding healthy tissues. In addition to this finding, in a study examining oral potentially malignant lesions (OPLs), it was found that a significant proportion of cases exhibited methylated RARβ promoters. Specifically, more than half of the cases, approximately 53%, showed methylation of the RARβ gene promoter [361]. 

On the whole, these findings raise uncertainty regarding the timing and significance of RARβ and RARβ2 methylation in the development of OSCC. It is unclear whether RARβ and RARβ2 methylation occurs as an early event in OSCC carcinogenesis or if it is a more widespread abnormality affecting various oral mucosal tissues.

##### RASSF

The RASSF (Ras-association domain family) gene family consists of ten members that play important roles in various cellular functions. The RASSF proteins function as tumor suppressors and play a role in governing various cellular processes, including the cell cycle, apoptosis, and the formation of microtubules [362]. In recent years, it has been recognized that the expression of several RASSF gene family members can be influenced by abnormal methylation of their respective genes. As an example, in one study investigating RASSF2, it was found that 39% of OSCC tissues showed methylation in at least one RASSF2 gene [363]. Another study reported methylation in 22% and 28% of OSCC tissues for the RASSF1A and RASSF2A genes, respectively. Of interest, the concurrent methylation of both the RASSF1A and RASSF2A genes was linked to a lower likelihood of disease-free survival [364]; thus, their combined impact on the progression of OSCC may be more pronounced. Further studies have corroborated these results, revealing methylation levels between 12% and 38% for the RASSF genes in OSCC tissues [261,309,363,364]. Moreover, in a study specifically investigating OSCC cases associated with the consumption of betel nuts, a cultural habit known to elevate the risk of oral cancer, 93% of the OSCC tissues exhibited methylation of the RASSF1 gene [365]. In another study utilizing PCR-based techniques to examine the methylation and mutation status of several genes in OSCC samples, the results revealed that the activated Ras/PI3K/AKT pathway have a stronger impact on survival in patients who underwent radiotherapy after surgery, and methylation of RASSF1A/RASSF2A is the most common mechanism that can be observed in OSCC cases treated with radiotherapy. Thus, these findings suggest that the epigenetic silencing of TSGs involved in the Ras/PI3K/AKT pathway plays a crucial role in OSCC radioresistance [364].

In short, the methylation-induced silencing of RASSF genes through epigenetic mechanisms may have a significant impact on both the development and prognosis of patients with OSCC.

### 3.2. Histone Modifications in OSCC

Histone modifications refer to chemical alterations that occur on histone proteins, which are the proteins around which DNA is wrapped to form a complex structure called chromatin. Histones can undergo several types of modifications, including methylation, acetylation, ubiquitination, phosphorylation, sumoylation, and others. These modifications can occur at specific amino acid residues within the histone proteins, such as lysine, arginine, serine, threonine, and others. Histone modifications can either activate or repress gene expression by affecting the accessibility of DNA to transcriptional machinery. For instance, acetylation promotes gene activation by loosening histone–DNA interactions, while methylation can have activating or repressing effects depending on the specific amino acid involved [366]. 

A study investigated the relationship between DNA and histone methylation patterns in normal, OPL and OSCC tissues. The findings revealed a positive correlation between DNA and histone methylation, suggesting that changes in DNA methylation were accompanied by alterations in histone methylation [367]. In relation to OSCC malignancy, the specific pattern of H3K4 histone methylation was found to be significant. OSCC tissues displayed higher levels of H3K4me2 histones, which are transcriptionally inactive, whereas normal tissues had more H3K4me3 modifications associated with gene activation. Similar histone methylation patterns were observed in OPLs, particularly in leukoplakia, suggesting that leukoplakia should be considered a premalignant condition for OSCC. These findings indicate that changes in H3K4 histone methylation may occur early in the development of OSCC [283].

Accordingly, the activation of gene transcription in OSCC was associated with the hypomethylation of the H3K9 histone, which resulted in the formation of loosely structured euchromatin. Indeed, when an OSCC cell line was treated with ornithine decarboxylase antizyme-1, it caused a reduction in H3K9me2 methylation levels and also influenced the expression of several DMNTs, indicating that the hypomethylation of histones plays a significant role in accelerating OSCC tumorigenesis [368].

Histone deacetylation, which is controlled by various enzymes called HDACs, plays a significant role in the development of oral cancer. One specific HDAC, HDAC2, has been found to enhance the stability of the HIF-1α protein, leading to increased invasion and migration in OSCC [369]. Another HDAC, HDAC6, was found to be upregulated in OSCC, and its activity was shown to be stage-specific, with higher activity observed in advanced stages of the disease [370]. The researchers concluded that the expression of HDAC6 may be crucial in determining the aggressiveness of tumors in oral cancer. Furthermore, HDAC6 was found to not only deacetylate histones but also deacetylate α-tubulin, thereby promoting cell motility dependent on microtubules, which is a process that may contribute to metastasis development [371]. 

Arif et al. made an interesting discovery regarding histone H3, specifically the H3K14 modification, in OSCC. They found that H3, particularly H3K14, showed increased acetylation levels in OSCC. The researchers also demonstrated that this enhanced H3 acetylation in the KB oral cancer cell line was dependent on nitric oxide. Since nitric oxide is produced during the inflammatory process, which is associated with OSCC initiation and development, this finding holds significant importance. Additionally, a potential therapeutic drug called hydrazinocurcumin showed the ability to inhibit histone acetyltransferase activity and reduce oral tumor growth in a mouse model. These experimental results provide mechanistic insights into the role of H3K14 hyperacetylation in the pathogenesis of OSCC [372]. 

The PARP enzyme family is responsible for adding ADP-ribose to proteins and forming poly (ADP-ribose) polymers. In OSCC, PARP-1 activity, DNA synthesis rates, and ADP-ribosylation levels are highest during active cell proliferation, particularly affecting histones [373]. The addition of poly (ADP-ribose) to histones leads to a loosening of the chromatin structure, which aids in DNA repair through protein complexes like chromatin assembly factor (CAF)-1. CAF-1 integrates acetylated histones (H3K56) into the chromatin and is upregulated in the nucleus of OSCC, indicating a poor prognosis and increased metastatic behavior [374,375]. Similar deregulation of cell proliferation and DNA repair involving CAF-1 is observed in aggressive tongue squamous cell carcinoma (TSCC). Immunohistochemical analyses have shown that CAF-1/p60 is commonly expressed in TSCC tissues, while CAF-1/p150 may be downregulated. Both of these factors are associated with unfavorable clinical outcomes and prognosis [375,376]. In addition, in OSCC clinical samples, there were observed changes in the levels of two histone markers, namely H3K4ac and H3K27me3. H3K4ac showed a decrease, while H3K27me3 exhibited higher levels. These alterations in histone modifications were found to be associated with advanced stages of OSCC, as well as cancer-specific survival and disease-free survival. The authors of the study suggest that this specific pattern of histone modifications could serve as a valuable prognostic tool in OSCC, providing important information about the progression and outcome of the disease [377].

Unlike extensive research conducted on other histone modifications like methylation and phosphorylation in various cancer types, the role of acetylation in the development of OSCC has been relatively understudied. The process of histone acetylation is regulated by a balance between enzymes called histone acetyltransferases (HATs) and HDACs. In the specific case of OSCC, several HDACs have been found to be associated with changes in differentiation, cell cycle-related genes, programmed cell death, angiogenesis, and the spread of cancer cells [378,379].

The evidence has suggested that in OSCC, there are specific changes in the levels of two histone deacetylases, namely HDAC1 [380] and HDAC2 [381]. HDAC1 is overexpressed in OSCC and contributes to its growth and progression by regulating a particular signaling pathway called miR-154-5p/PCNA. This pathway affects cellular processes related to proliferation and DNA replication, which are important for cancer advancement [382]. Similarly, HDAC2 shows increased mRNA and protein levels in OSCC and premalignant lesions compared to normal tissue. The higher expression of HDAC2 indicates its potential involvement in the development of OSCC. Moreover, the protein level of HDAC2 is associated with the histological differentiation of cancer cells and the tumor-node-metastasis stage, which describes the extent and spread of the tumor. This emphasizes the importance of HDAC2 in the transition from precancerous to malignant forms of OSCC [381].

Nicotinamide N-methyltransferase (NNMT) is an enzyme that plays a role in epigenetics, the study of heritable changes in gene expression without alterations to the DNA sequence [383]. NNMT influences epigenetics by interacting with and modulating the activity of enzymes involved in this regulatory process, such as HDACs and sirtuins [384,385,386]. In the context of cancer, NNMT has been found to be overexpressed in various solid tumors, including OSCC [225,387]. This increased expression of NNMT in cancer cells has been associated with enhanced tumorigenicity and aggressiveness, suggesting that NNMT may have a role in promoting cancer progression [387,388,389]. Sirtuins, a family of NAD-dependent enzymes, are crucial in cellular processes like gene expression, DNA repair, cell cycle progression, metabolism, and longevity in humans [390]. NNMT affects sirtuins’ activity by affecting NAD levels. NNMT converts nicotinamide to 1-MNA, consuming NAD and reducing its availability. This decrease affects sirtuins’ enzymatic activity, which regulates target proteins like transcription factors and metabolic pathways through deacetylation [391]. When NAD levels are limited due to NNMT-mediated consumption, the activity of sirtuins may be impaired. Reduced sirtuin activity can disrupt their ability to perform deacetylation and regulate cellular processes, thereby causing cancer cell progression [392,393]. 

Chromatin remodelers play a crucial role in OSCC by modulating the positioning of nucleosomes and the accessibility of chromatin in response to DNA-related stimuli [394]. One example of a genome-organizing protein is special AT-rich sequence binding protein 1 (SATB1), which can modify the structure of chromatin and direct chromatin remodeling enzymes to specific regions to regulate gene expression. Increased levels of SATB1 expression have been linked with metastasis, poor prognosis, and decreased survival rates [395] in head and neck squamous cell carcinoma (HNSCC) [396].

Another protein, zinc finger and SCAN domain-containing 4 (ZSCAN4), can alter the epigenetic profile and chromatin state in tumors [397]. ZSCAN4 promotes the hyperacetylation of histone 3 at specific gene promoters, such as OCT3/4 and NANOG, leading to the upregulation of factors associated with cancer stem cells (CSCs) [398]. In contrast, depletion of ZSCAN4 results in the downregulation of CSC markers, reduced ability to form tumorspheres, and a limitation in tumor growth [399]. This highlights the importance of ZSCAN4 in maintaining the CSC phenotype and driving tumor progression in HNSCC [398].

In addition, a chromatin remodeler called RSF-1, which belongs to the ISWI family [400], is upregulated in OSCC and has been linked to increased invasion, lymph node metastasis, and advanced stages of carcinomas. RSF-1 is also associated with enhanced resistance of OSCC cells to both radiotherapy and chemotherapy, making it a significant factor in treatment resistance [401].

### 3.3. Non-Coding RNAs

Non-coding RNAs (ncRNAs) are RNA molecules that regulate gene expression and cellular processes, contributing to various biological functions and disease mechanisms [402]. There are two main types: small ncRNAs (miRNAs) and long ncRNAs (lncRNAs). Small ncRNAs, like microRNAs and small interfering RNAs, are post-transcriptional regulators, involved in development, cell proliferation, and differentiation. They have been shown to regulate gene expression at multiple levels and are associated with diseases like cancers [325,403,404]. Their study opens new avenues for understanding gene regulation and cellular processes. A total of 109 miRNAs were discovered to be differently expressed solely in progressive leukoplakia and invasive OSCC when the miRNA profiles of leucoplakia with and without advancement were compared [405]. Table 2 summarizes several miRNAs whose expression is deregulated in OSCC.

In addition to this, in a study, it was observed that in OSCC cell lines, 54 miRNAs (representing 37% of the miRNAs studied) were expressed at lower levels compared to normal cells. Furthermore, the researchers found that the reduced expression of miR-137, miR-34b, miR-203, and miR-193a in OSCC cell lines was associated with a process called CpG hypermethylation, not to mention that CpG hypermethylation involves the addition of methyl groups to specific DNA sequences (CpG sites) in gene promoter regions, which can lead to gene silencing or decreased gene expression. To confirm their findings, the researchers also examined OSCC tissues taken from patients. They observed that the downregulation of miR-137 and miR-193a through epigenetic silencing, likely CpG hypermethylation, was more frequently observed in these tissue samples than the reduced expression of miR-34b and miR-203. This suggests that the epigenetic silencing of miR-137 and miR-193a may play a significant role in the development of OSCC in living organisms [285].

In another study using an OSCC animal model, researchers conducted an experiment on six Syrian hamsters. These hamsters were treated with a carcinogen called DMBA (dimethylbenz[a]anthracene), and then the expression profiles of miRNAs were analyzed using a technique called miRNA microarray. The results of the analysis showed that five miRNAs (miR-200b, miR-21, miR-221, miR-762, and miR-338) were expressed at higher levels compared to normal, while twelve miRNAs (miR-26a, miR-16, miR-29a, miR-124a, miR-126-5p, miR-125b, miR-143, miR-148b, miR-145, miR-155, miR-203, and miR-199a) were expressed at lower levels. These changes in miRNA expression were a response to the oncogenic stimulus caused by the DMBA carcinogen [406]. To put it simply, it was observed that certain miRNAs are either overexpressed (higher levels) or downregulated (lower levels) in response to the carcinogen DMBA.

In OSCC, miR-211 is often overexpressed and associated with tumor progression, metastasis, invasion, and poor prognosis [407]. On the other hand, miR-133a and miR-133b are downregulated in OSCC [408], promoting cancer development by enhancing cell proliferation and inhibiting apoptosis. The underlying mechanisms for the differential regulation of these miRNAs are not well understood, but studies suggest that aberrant expression of the dicer enzyme, responsible for miRNA maturation, and potential involvement of the let-7 miRNA family may play a role [409].

According to a study examining the role of miRNA in OSCC, it was found that certain miRNAs, including miR21, miR181b, and miR345, are consistently highly expressed in progressive lesions and invasive OSCC, which may be involved in the development thereof [405]. The miR-17-92 cluster is a group of microRNAs that are transcribed together as a single polycistronic unit. Research has revealed that miR-17-92 can induce various biological processes, including both normal development and cancer development. Specifically, research findings have indicated that the expression levels of the miR-17-92 cluster are higher in cultured carcinoma cell lines when compared to normal human keratinocytes [410,411].

The other study aimed to investigate the expression levels of miR-133a and miR-133b in OSCC and normal epithelial tissues, as well as their involvement in cancer development. The results showed that both miRNAs were significantly decreased in OSCC samples compared to normal samples. Further analysis revealed that the decrease in miR-133a and miR-133b led to the activation of an oncogene called pyruvate kinase type M2. This oncogene is involved in several cellular processes that promote cancer progression, including the Warburg effect, a metabolic phenomenon where cancer cells switch to anaerobic metabolism [408].

Kozaki et al. conducted a study examining the changes in expression levels of miR-137 and miR-193a in certain OSCC cell lines. Their findings suggested that the epigenetic silencing of these miRNAs, resulting from DNA hypermethylation, may play a significant role in the progression of oral cancer [285].

## 4. Therapeutic Targeting of Epigenetic Mechanisms in OSCC

As mentioned above, epigenetic processes like DNA methylation, histone modifications, chromatin remodeling, and non-coding RNAs have a significant impact on OSCC. Targeting these epigenetic alterations has become a promising approach for therapy. This means that inhibiting enzymes like DNMTs and HDACs has shown promise in reversing epigenetic changes and reactivating TSGs [412]. Indeed, clinical trials have provided positive results with these inhibitors in treating OSCC. Additionally, there is ongoing exploration of other epigenetic targets such as chromatin remodeling complexes and non-coding RNAs. Despite such findings, further research is necessary to fully comprehend these epigenetic modifications and develop optimal therapeutic strategies for OSCC, and in the ensuing paragraphs, some important findings regarding this topic will be discussed. 

### 4.1. DNA Methylation Inhibitors as Therapeutic Agents in OSCC

CpG methylation alterations, which involve the addition or removal of methyl groups to specific regions of DNA, hold potential as targets for epigenetic therapies in OSCC. An example of this is the use of a DNMT inhibitor called zebularine. In a study involving OSCC cells (HSC-3), treatment with zebularine alone resulted in inhibition of tumor growth. This was evident through decreased cell growth and a reduction in the number of cells in the G2/M phase of the cell cycle [413]. The exact mechanisms underlying these observations are not yet fully understood. However, it is noteworthy that when zebularine was used in combination with chemotherapy drugs such as cisplatin or 5-fluoro-uracil, interesting effects were observed. Zebularine significantly enhanced the apoptotic activity induced by cisplatin treatment, potentially making the chemotherapy more effective in inducing cancer cell death. However, when zebularine was combined with 5-fluorouracil, it appeared to decrease the efficacy of the 5-fluorouracil treatment [414].

As previously mentioned, the MGMT gene is implicated in OSCC, and its expression is influenced by abnormal methylation. In OSCC cell lines, those exhibiting lower levels of MGMT methylation displayed a more pronounced response to 5-fluorouracil chemotherapy compared to cell lines with higher levels of MGMT methylation. However, when cells were treated with the MGMT inhibitor O6-benzylguanine (O6-BG), the non-responsive cells exhibited a significantly enhanced anti-proliferative effect in response to 5-fluorouracil. Thus, caution is necessary when considering combination therapies and inhibitors for OSCC treatment, as the methylation status of genes like MGMT can influence the effectiveness of chemotherapy [415]. 

Further investigation has been conducted on DNMT inhibitors such as 5-azacytidine (5-aza-CR) and 5-aza-2′-deoxycytidine (5-aza-CdR or decitabine) due to their capacity to inhibit DNA methyltransferases and induce genomic hypomethylation. Notably, 5-aza-CdR has demonstrated the ability to reduce methylation levels in cells. In the context of OSCC, studies have revealed that these compounds can reactivate the expression of the p16^INK4a^ gene [416]. As a result, the use of demethylating agents like 5-aza-CdR can potentially reverse this methylation and restore the expression of p16^INK4a^ in OSCC cells.

#### Natural Compounds as DNA Methylation Inhibitors in OSCC

Green tea extracts have shown promise in preventing the progression of precancerous lesions to OSCC. Clinical studies have demonstrated that patients treated with green tea extracts had a better response rate compared to those who received a placebo. Higher doses of green tea extracts also resulted in a significantly higher response rate compared to the placebo group [417]. Similarly, in a long-term study involving a large cohort of individuals—a total of 20,550 men and 29,671 women aged 40–79 years—without a history of oral cancer, an inverse relationship was observed between green tea consumption and the risk of oral cancer, particularly in women. Accordingly, women who had a higher intake of green tea (ranging from 1–2 cups, 3–4 cups, or 5 cups per day) exhibited a reduced risk of developing oral cancer compared to those who consumed only 1 cup per day. However, in men, the correlation was less pronounced and did not achieve statistical significance due to the limited number of cancer cases included in the study [418].

Epigallocatechin-3-gallate (EGCG), a primary component of green tea, has demonstrated significant efficacy in inhibiting the invasive capabilities of OSCC. It achieves this by reversing the hypermethylation process of the RECK gene, leading to an upregulation of RECK mRNA expression. In addition, EGCG treatment at a concentration of 50 μM has been observed to suppress the activity of MMP-2 and MMP-9 enzymes in these cells [419]. Furthermore, EGCG has been found to counteract the gene silencing effects caused by methylation in various genes, such as p16^INK4a^, RARβ, MGMT, and human mutL homologue 1 (hMLH1), in human esophageal cancer cells. This reactivation of genes was observed at a concentration of 20 μM, which is considered effective for stimulating the activity of these genes [420].

Genistein, a natural compound found in soybeans, exhibits significant effects on epigenetic processes in OSCC. It has the ability to suppress the activity of DNA methyltransferase, reverse DNA hypermethylation, and reactivate genes that were silenced by methylation. In OSCC, genistein has been demonstrated to reactivate key genes, including RARβ, p16^INK4a^, and MGMT, which play important roles in OSCC development. Moreover, genistein can hinder cell proliferation at specific concentrations, yet the impact of genistein on DNA methylation is dependent on dosage and influenced by the presence of substrates and methyl donors [421]. When compared to genistein, biochanin A and daidzein showed less effectiveness in inhibiting DNA methyltransferase activity, reactivating the RARβ gene [422], and suppressing cancer cell growth. However, when genistein was combined with trichostatin, sulforaphane, or 5-aza-dCyd, it was able to enhance the reactivation of these silenced genes and further inhibit cell growth. This indicates that genistein has the potential to work synergistically with other compounds to achieve a stronger reactivation of silenced genes and more effective inhibition of cancer cell growth [421]. 

### 4.2. HDAC Inhibitors as Therapeutic Agents in OSCC

The potential of histone modifications in human cancer has sparked interest in utilizing epigenetic inhibitors for the treatment of OSCC. Scientists are investigating various inhibitors that specifically target HDAC activity. HDACs are classified into four groups (Class I–IV) based on their function and sequence similarity, with a total of 18 different HDACs identified. These HDACs exhibit different affinities for distinct classes [423]. HDAC inhibitors represent a promising class of compounds that prevent histone deacetylation, leading to a more relaxed chromatin structure [424]. This activation allows for the regulation of genes involved in cell survival, proliferation, differentiation, and apoptosis. In OSCC, HDAC inhibitors have the potential to reactivate silenced TSGs, reversing malignant characteristics. Notably, combining HDAC inhibitors with established chemotherapeutic agents produces synergistic effects, enhancing the effectiveness of conventional chemotherapy [425].

Trichostatin A (TSA), an early HDAC inhibitor, selectively targets specific HDACs in OSCC and TSCC cell lines [426]. It hinders cell growth, causes cell cycle arrest, and triggers apoptosis by influencing the expression of key proteins, namely p21^WAF1^, Cyclins [427,428], Bax [428], and Bcl-2 family members. However, TSA’s potential as an anticancer treatment may be restricted due to its considerable toxicity [429].

Butyric acid derivatives, such as phenylbutyrate and sodium butyrate, are well-researched HDAC inhibitors. Phenylbutyrate promotes DNA repair and cell survival while reducing oxidative stress, TNF-α expression, and severe oral mucositis. It also lowers the risk of OSCC and tumor progression when used alongside radiotherapy [430]. Sodium butyrate inhibits OSCC cell proliferation and induces G1 and G2/M cell cycle arrest in several human OSCC cell lines, including HSC-3, HSC-4, SCC-1, and SCC-9 [431,432]. It modulates the expression of cell cycle regulatory proteins, increasing CDK6, p21^WAF1^, and p27 in the G1 phase while decreasing CDK2 and phosphorylation of Rb in the S/G2 phase. Sodium butyrate also regulates the expression of cyclins, upregulating Cyclin D1 and downregulating Cyclin B1 and Cyclin E [432]. Despite being considered weak HDAC inhibitors, phenylbutyrates and sodium butyrates are being studied in clinical trials, especially for their effects on myelodysplastic disorders [426].

Romidepsin, a potent inhibitor of Class I HDACs, effectively blocks the activity of these HDAC enzymes [426]. Similar to butyric acid derivatives, romidepsin causes a halt in the cell cycle at the G1 and G2/M phases [433]. In laboratory studies using OSCC cells, romidepsin treatment has been shown to induce a time-dependent re-expression of mapsin transcripts, which is a proposed tumor suppressor gene. This re-induction of mapsin expression after romidepsin treatment may play a role in restoring normal biological functions, including the prevention of cell invasion and tumor angiogenesis. Another interesting finding is that romidepsin treatment increases the expression of hTERT, a gene involved in regulating cell senescence [434,435]. Both mapsin and hTERT are important factors in controlling cell mortality and invasion. By targeting the re-expression of these genes, romidepsin, as an HDAC inhibitor, could have significant implications in cancer therapy. Alongside romidepsin, there is a wide range of HDAC inhibitors currently under development and evaluation for their potential as effective anti-tumor agents [436]. 

In another study, researchers investigated the effects of the HDAC inhibitor entinostat on OSCC cells. They found that entinostat significantly reduced cell proliferation, induced cell cycle arrest at the G0/G1 phase, and triggered substantial tumor cell apoptosis. The drug also increased the production of ROS within the cancer cells and led to a notable reduction in CSCs [437]. Additionally, entinostat caused changes in histone acetylation and the expression of cell cycle-associated proteins. Overall, these findings suggest that entinostat has the potential to be an effective therapeutic agent for OSCC by inhibiting tumor growth, inducing cell death, increasing oxidative stress, and targeting CSCs [438]. Likewise, in a study investigating the antitumor activity of apicidin, an HDAC inhibitor, in murine OSCC cells researchers found that apicidin inhibited cell proliferation and selectively reduced the expression of HDAC8 in OSCC cells. It induced apoptosis and autophagy in the treated cells. In a mouse model, apicidin significantly inhibited tumor growth and reduced cell proliferation while promoting apoptosis and autophagy in tumor tissues. These findings suggest that apicidin has potential as an effective therapeutic agent for OSCC [439].

## 5. Epigenetics behind Drug Resistance

Numerous studies have highlighted the importance of epigenetic changes in cancer drug-tolerant persister (DTP) cells, which are a subset of cells that survive within a heterogeneous population and develop increased tolerance to drug treatments [440,441,442]. These epigenetic changes contribute to the survival and maintenance of CSCs, which are known to be resistant to therapies [443]. Cancer cells can modify their epigenomic landscape through processes such as DNA methylation and modifications of histone and non-histone proteins, enabling them to develop various mechanisms of drug resistance as a means to evade anti-cancer therapies. Epigenetic regulators, including DNMTs, chromatin readers, writers, and erasers, as well as different histone modifiers and non-coding RNAs, play a role in fine-tuning the regulation of genes involved in multiple forms of therapy resistance [444]. Overall, these epigenetic changes and regulatory mechanisms contribute to the development of drug resistance in cancer cells, posing a significant challenge in the treatment of cancer.

### 5.1. DNA Methylation and Drug Resistance

Little has been reported on the relationship between DNA methylation and drug resistance in OSCC cases, yet several studies have established a strong link between DNMTs and the development of resistance in other tumors. DNMT1, in particular, can regulate CSCs and their resistance mechanisms. For example, in liver stem cells, DNMT1 controls the activation of Wnt/β-catenin signaling through a protein called BEX1, which helps maintain the self-renewal capacity of these cells [445]. In glioma stem cells (GSCs), DNMT1 interacts with CD133, a marker of CSCs, to prevent its own nuclear translocation. This interaction preserves the self-renewal and tumorigenic properties of GSCs, making them resistant to the chemotherapy drug temozolomide [446]. Additionally, DNMT1-induced hypermethylation of the promoter region of miR-34a leads to its inactivation and abnormal activation of the Notch pathway. In pancreatic cancer cells, treatment with decitabine, an inhibitor of DNMT1, enhances their sensitivity to the drug sorafenib [447]. The overexpression of DNMT3A and DNMT3B has been observed in rhabdomyosarcoma and tamoxifen-resistant breast cancers, indicating a potential target for improving the effectiveness of radiation and chemotherapy [448,449]. Similarly, the increased expression of DNMT3B is associated with sorafenib resistance in hepatocellular carcinoma cells, and inhibiting DNMT3B can increase their sensitivity to sorafenib [450,451]. However, conflicting findings suggest that a high expression of DNMTs may actually make certain types of cancers more sensitive to decitabine treatment, such as triple-negative breast cancers [452] and ovarian cancers [453]. Overall, these studies reveal that DNMTs have a dual role in tumor resistance, and their impact varies depending on the specific type of cancer and its context.

DNA demethylases, specifically ten-eleven translocations (TETs), play a significant role in drug resistance mechanisms [454]. TET1, for instance, is responsible for the oxidation of methyl cytosine to hydroxymethyl cytosine, which promotes EMT and increases ROS levels [455,456]. In gastric cancer cells, TET1 can induce resistance to the chemotherapy drug oxaliplatin [457], and it has also been implicated in inducing resistance in lung cancer cells [458]. The role of TET2 in tumor resistance is not well studied, but the upregulation of TET3 has been shown to increase the expression of tumor suppressor genes, leading to the inhibition of growth and self-renewal ability in glioblastoma stem cells [459]. 

### 5.2. Histone Modifications and Drug Resistance

Histone acetylation, regulated by HATs, contributes to cancer development by affecting gene transcription and chromatin structure. The imbalance between HATs and HDACs can lead to tumor formation [460]. Various HATs, including MYST1, TIP60, CBP/p300, and GCN5, are abnormally expressed in different cancers [461,462,463]. HATs have been implicated in drug resistance, with high expression of HAT1 associated with resistance to certain drugs in prostate and pancreatic cancer. Inhibition of KAT6A enhances sensitivity to cisplatin in ovarian cancer, while KAT2A contributes to tamoxifen resistance in breast cancer. P300 has a dual role in drug resistance, as its loss can mediate resistance to PORCN inhibition in pancreatic cancer, while its upregulation can promote resistance to apatinib in gastric cancer and BRAF inhibitors in melanoma [444]. Targeting HATs may provide a strategy to overcome drug resistance in cancer treatment.

Notably, a high expression of HDACs, particularly HDAC6, can contribute to tumor resistance. In specific cancers like non-small cell lung cancers and melanoma, HDAC6 plays a crucial role by increasing the stability of proteins like EGFR and tubulin β3 [464,465]. This stabilizing effect reduces apoptosis and promotes tumor resistance. Additionally, SIRT7 and SIRT1, members of the sirtuin family of HDACs, have been implicated in increasing resistance to the chemotherapy drug cisplatin [444,466,467]. NNMT inhibitors hold promise as targeted therapies for cancer. As mentioned above, NNMT affects NAD homeostasis, leading to decreased NAD+ levels, which can impact NAD+-dependent enzymes like SIRTs. Inhibiting NNMT activity can restore NAD+ levels, modulate the availability of SAM (S-adenosylmethionine), and potentially influence DNA and histone methylation patterns. By restoring NAD+ levels and modulating epigenetic processes, NNMT inhibitors have the potential to restore normal gene expression patterns and counteract the aberrant epigenetic changes observed in cancer cells. NNMT inhibitors can be beneficial therapeutic agents by restoring NAD+ levels, modulating SAM availability, and potentially influencing DNA and histone methylation patterns. These benefits can enhance cellular metabolism and energy production, and potentially counteract aberrant epigenetic changes associated with cancer development. However, further research is needed to fully understand the efficacy, safety, and potential side effects of these inhibitors, and clinical trials are needed to determine optimal use conditions [388,389,468,469].

Targeting HDACs has emerged as a potential strategy to limit the development of drug resistance in cancer. HDAC inhibitors have been developed and four of them have received approval from the FDA for cancer treatment. Furthermore, researchers are exploring the combination of HDAC inhibitors with other targeted therapies to enhance their effectiveness in overcoming drug resistance [444].

## 6. Conclusions

As mentioned above in detail, epigenetic modifications such as DNA methylation and histone modifications play crucial roles in OSCC. Aberrant DNA methylation of TSGs leads to gene silencing and promotes tumor progression. Specific gene methylation patterns have potential as diagnostic biomarkers for OSCC. Epigenetic modifiers like DNMT inhibitors and HDAC inhibitors can modify abnormal epigenetic patterns, reactivating TSGs and suppressing oncogenes because these targeted therapies directly alter gene expression and minimize side effects. miRNAs also contribute to OSCC pathogenesis and treatment, paving a new way in diagnosing and preventing OSCC. Notably, epigenetic mechanisms play a significant role in drug resistance in OSCC and other cancers. For instance, aberrant DNA methylation can silence tumor suppressor genes, while alterations in histone modifications and chromatin remodeling affect gene expression patterns related to drug metabolism and apoptosis. Thus, gaining knowledge about and focusing on these epigenetic mechanisms provide promising approaches to address drug resistance and enhance the effectiveness of cancer treatments. As per such findings, understanding the interplay between epigenetic modifications and OSCC holds promise for improving diagnostics and guiding targeted therapeutic approaches, and to this end, further research in this field is necessary for advancing OSCC management.

## Figures and Tables

**Figure 1 cancers-15-05600-f001:**
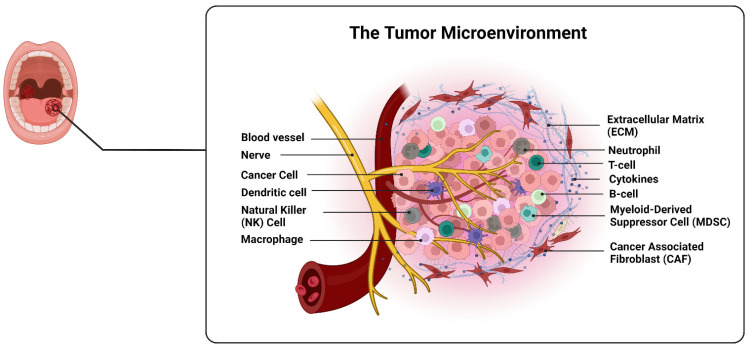
The TME refers to the complex cellular and non-cellular components surrounding a tumor, which interact and influence tumor growth, progression, and response to treatment. It consists of various cell types, including tumor cells, immune cells, fibroblasts, endothelial cells, and adipocytes, as well as extracellular matrix components, cytokines, growth factors, and signaling molecules. The TME plays a critical role in shaping tumor behavior by promoting angiogenesis, facilitating immune evasion, inducing tissue remodeling, and providing a supportive niche for tumor cells [78].

**Figure 2 cancers-15-05600-f002:**
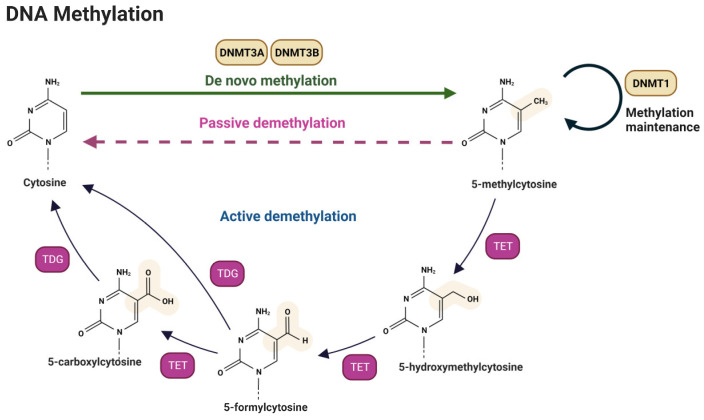
DNA methylation is a crucial biochemical process that adds or removes a methyl group from the DNA molecule and plays a crucial role in gene regulation, genomic stability, and cellular differentiation. It involves two main processes: de novo methylation, which establishes new DNA methylation patterns during early development or in specific cell types, and demethylation, which removes methyl groups from the DNA molecule. De novo methylation is essential for embryonic development, tissue-specific gene expression, and silencing of repetitive DNA elements. Demethylation can occur through active or passive demethylation, with active demethylation involving enzymatic processes such as oxidation and base excision (one prominent pathway involves the Ten-Eleven Translocation (TET) family of enzymes), while passive demethylation occurs through DNA replication without de novo methylation. Both processes play essential roles in the regulation of gene expression and cellular differentiation [87].

**Figure 3 cancers-15-05600-f003:**
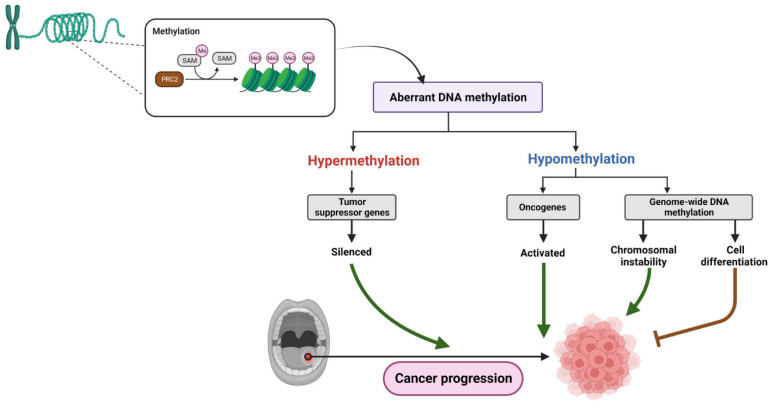
Aberrant DNA methylation is a key contributor to tumorigenesis and tumor formation and growth. Hypermethylation of tumor suppressor gene promoters leads to their silencing, impairing their ability to regulate cell growth and prevent tumor formation. Conversely, hypomethylation of oncogene promoters activates these genes, promoting abnormal cell growth and tumor development. Genome-wide DNA hypomethylation, which reduces the methylation levels throughout the genome, can have dual effects. It may induce chromosomal instability, increase the likelihood of genetic mutations, and promote tumorigenesis. Hypomethylation can also drive cell differentiation and inhibit tumorigenesis by promoting normal cell development and function. These aberrant DNA methylation patterns play crucial roles in the complex mechanisms underlying cancer development.

**Figure 4 cancers-15-05600-f004:**
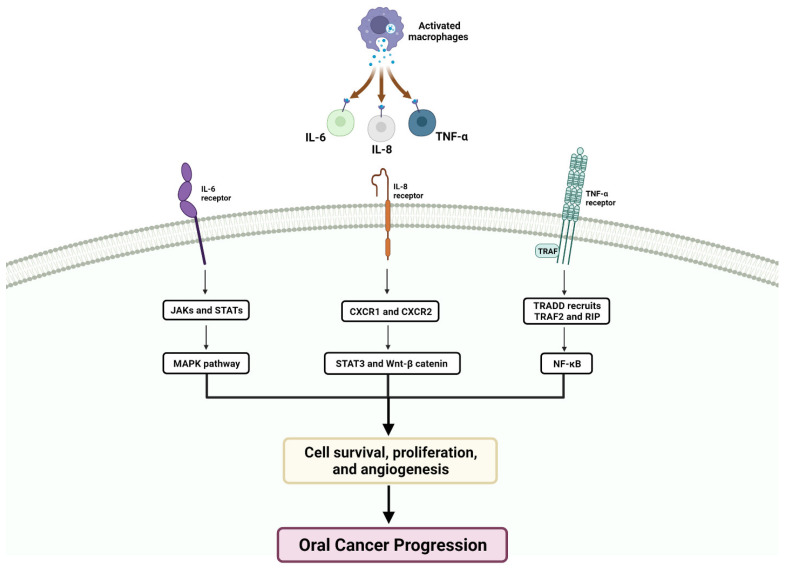
The schematic diagram illustrates how IL-8, IL-6, and TNF-α are activated by macrophages in the tumor microenvironment. Upon activation, they initiate signaling pathways that affect various aspects of the tumor microenvironment, including inflammation, cell proliferation, and angiogenesis, which lead to cancer progression.

**Figure 5 cancers-15-05600-f005:**
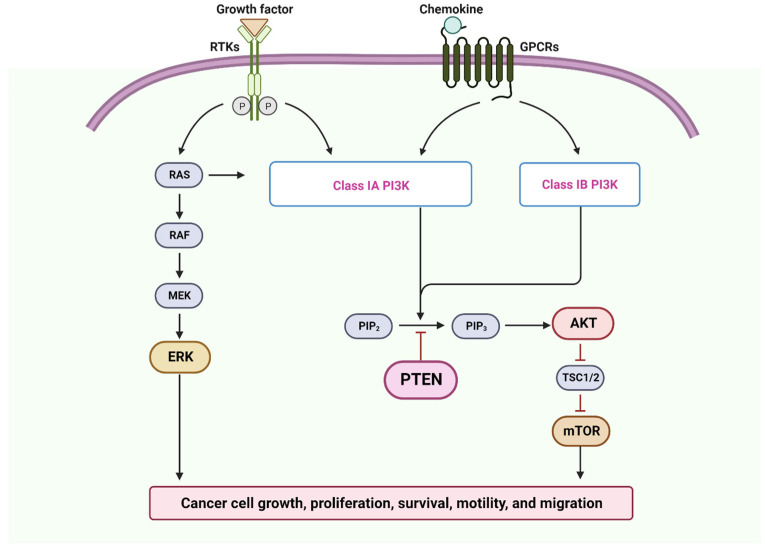
The PTEN tumor suppressor network is a vital part of the PI3K-Akt pathway, which regulates cell growth and survival. PTEN inhibits the pathway by reducing the levels of a signaling molecule called PIP3. When PTEN is inactive or lost in cancer, the PI3K-Akt pathway becomes overactive. This leads to increased activity of mTOR, a protein that controls cell growth and metabolism. Dysregulation of mTOR contributes to tumor progression and resistance to cancer therapies. Understanding this network is important for developing targeted treatments to restore pathway balance and inhibit cancer growth. Furthermore, dysregulated RAS-ERK pathways, triggered by mutations in RAS genes, promote uncontrolled cell growth. Growth factors in tumor microenvironments further stimulate this pathway, making targeting it a major focus in cancer research for therapeutic strategies [323,324,328].

**Table 1 cancers-15-05600-t001:** Most common hypermethylated genes.

Gene	Locus	Function	Refs.
*ABO*	9q34	Blood group antigen	[271]
*APC*	5q21	Signal transduction	[272,273]
*ATM*	11q22-q23	Tumor suppressor	[274,275,276]
*C/EBPα*	19q13	Tumor suppressor	[277]
*CDKN2A*	9p21	Cell cycle	[274]
*CRABP2*	1q21	Nuclear transcriptional regulator	[278]
*DAPK*	9q	Apoptosis	[28,271,274,275,279]
*DCC*	18q21	Tumor suppressor	[28,271,274,275,279]
*DKK3*	11p	Transcriptional regulator	[280]
*E-cadherin*	16q22	Signal transduction	[28,32]
*EDNRB*	13q22	Signal transduction	[281]
*GSTP1*	11q13	Detoxification of carcinogens	[275,282]
*H3K4*	1q21.2	Histone	[283]
*HIN1*	12p13	Tumor suppressor	[284]
*Hmlh1*	3p21	DNA repair	[275]
*LHX6*	9q33	Transcriptional regulator	[277]
*MGMT*	10q26	DNA repair	[275]
*MINT family*	/	/	[275]
*miR137*	1p21.3	Tumor suppressor	[285]
*miR193a*	17q11.2	Tumor suppressor	[285]
*MX1*	21q22	/	[278]
*p14*	9p21	Apoptosis	[275]
*p15*	9p21	Cell cycle	[279]
*p16*	9p21	Cell cycle	[275]
*p53*	17p13	Tumor suppressor	[275]
*p73*	1p36	Apoptosis	[275]
*PTEN*	10q23	Tumor suppressor	[286]
*RARB2*	17q21	Nuclear transcriptional Regulator	[275]
*RASSF-1*	3p21	Apoptosis	[279]
*Rb*	13q14	Tumor suppressor	[273]
*RUNX3*	1p36	Transcriptional regulator	[281]
*SFRP1*	8p11.21	Transcriptional regulator	[287]
*SFRP1-2-4-5*	8p11.21, 4q31.3, 7p14.1, 10q24.1	Transcriptional regulator	[287,288]
*TCF21*	6q23-q24	Epithelial–mesenchymal interactions	[277]
*THBS1*	15q15	cell-to-cell and cell-to-matrix interactions	[273]
*TIMP3*	22q12	Epithelial–mesenchymal interactions	[279]
*WIF1*	12q14	Transcriptional regulator	[287]
*σ-14-3-3*	1p36	Signal transduction	[275]

**Table 2 cancers-15-05600-t002:** miRNAs whose expression is dysregulated in OSCC.

microRNAs	Expression in OSCC	Cellular Function
miR-193a/miR-137 miR-503/miR-133b/miR-15a/miR-133a	↓	Proliferation and apoptosis
miR-184/miR-24/miR-21	↑	Proliferation and apoptosis
miR-138/miR-222	↓	Metastasis
miR-31/miR-211	↑	Metastasis
miR-21	↓	Chemoresistance
miR-98/miR-23a/miR-214	↑	Chemoresistance

↑: Upregulated; ↓: Downregulated.

## Data Availability

All data and materials are within the paper.

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
