# Peer review of "Epigenetic Regulation in Oral Squamous Cell Carcinoma Microenvironment: A Comprehensive Review"

_cancers, 2023, doi:10.3390/cancers15235600_

Round 1

Reviewer 1 Report

Comments and Suggestions for Authors

The manuscript “Epigenetic Regulation in Oral Squamous Cell Carcinoma Microenvironment: A Comprehensive Review” is a review article regarding the current state-of-art of epigenetic alterations in oral squamous cell carcinoma and its tumor microenvironment.

The manuscript provides a decent view on the topic and may be of interest for the readers. I appreciate the long and detailed work conducted by authors. Nonetheless, there are important concerns that needs to be addressed by authors in order to consider the manuscript suitable for publication:

1.    I do not understand why the introduction section seems to be excluded from the main text, while it should be part of it.

2.    The introduction section should include more information about oral squamous cell carcinoma epidemiology.

3.    The quality of figure 1 is poor with consequent difficulties of reading and therefore should be improved.

4.    The main concern is that this review completely ignores a major regulator of cell and tumor microenvironment epigenetics, the enzyme nicotinamide N-methyltransferase (NNMT) which can influence a number of enzymes involved in epigenetics, such as histone deacetylases sirtuins (PMID: 36829935; PMID: 36817086; PMID: 31043742).

This enzyme has been reported to be overexpressed in a variety of solid tumors including oral squamous cell carcinoma, where it contributes to the tumorigenicity and aggressiveness (PMID: 34827592).

Since NNMT can affect NAD homeostasis, NAD-dependent enzymes and concentration of SAM, it has a great impact on methylation status of genome and epigenetics, as demonstrated by Ulanovskaya et al. (PMID: 23455543). Notably, a number of NNMT inhibitors are already available, and their use is a promising strategy for targeted therapy in cancer (PMID: 34572571; PMID: 34704059; PMID: 34424711; PMID: 36104373).  All these considerations should be included since they would let the manuscript cover totally the topic of possible epigenetic targets in oral squamous cell carcinoma.

5.    Moreover, in the paragraph 3.1.1.8. authors ignored that a positive correlation between NNMT and survivin ΔEx3 isoform expression levels was reported in Oral Squamous Cell Carcinoma tissue samples. (PMID: 29882109).

6.    Table 2 reports miRNA which are upregulated or downregulated. Therefore in the title of the table the word “deregulated” should be changed with “dysregulated”. Moreover, in the table I also suggest replacing the words “upregulated” and “downregulated” with arrow up and arrow down symbols.

Comments on the Quality of English Language

Moderate editing of English language required

Author Response

1-The manuscript “Epigenetic Regulation in Oral Squamous Cell Carcinoma Microenvironment: A Comprehensive Review” is a review article regarding the current state-of-art of epigenetic alterations in oral squamous cell carcinoma and its tumor microenvironment.

The manuscript provides a decent view on the topic and may be of interest for the readers. I appreciate the long and detailed work conducted by authors. Nonetheless, there are important concerns that needs to be addressed by authors in order to consider the manuscript suitable for publication:

  1. I do not understand why the introduction section seems to be excluded from the main text, while it should be part of it.

Response: Thank you for your suggestion. It has been addressed.

  1. The introduction section should include more information about oral squamous cell carcinoma epidemiology.

Response: Thank you for your insightful recommendation. One paragraph has been dedicated to this part, so it has been addressed. 

  1. The quality of figure 1 is poor with consequent difficulties of reading and therefore should be improved.

Response: The quality of the figure 1 have been improved.

  1. The main concern is that this review completely ignores a major regulator of cell and tumor microenvironment epigenetics, the enzyme nicotinamide N-methyltransferase (NNMT) which can influence a number of enzymes involved in epigenetics, such as histone deacetylases sirtuins (PMID: 36829935; PMID: 36817086; PMID: 31043742).

This enzyme has been reported to be overexpressed in a variety of solid tumors including oral squamous cell carcinoma, where it contributes to the tumorigenicity and aggressiveness (PMID: 34827592).

Since NNMT can affect NAD homeostasis, NAD-dependent enzymes and concentration of SAM, it has a great impact on methylation status of genome and epigenetics, as demonstrated by Ulanovskaya et al. (PMID: 23455543). Notably, a number of NNMT inhibitors are already available, and their use is a promising strategy for targeted therapy in cancer (PMID: 34572571; PMID: 34704059; PMID: 34424711; PMID: 36104373).  All these considerations should be included since they would let the manuscript cover totally the topic of possible epigenetic targets in oral squamous cell carcinoma.

Response: Thank you for your insightful suggestions and informative note that you mentioned. It was useful to improve this manuscript. We have addressed these recommendations. We have concisely mentioned new information regarding the effects of NNMT on histone modification and the mechanisms by which it affects cancer cells (subheading 3.2.). Further, we have concisely added information about NNMT inhibitors and their mechanisms as promising targeted therapy in cancer (subheading 5.2.).  

  1. Moreover, in the paragraph 3.1.1.8. authors ignored that a positive correlation between NNMT and survivin ΔEx3 isoform expression levels was reported in Oral Squamous Cell Carcinoma tissue samples. (PMID: 29882109).

Response: Thank you for your insightful recommendation. It has been addressed.

  1. Table 2 reports miRNA which are upregulated or downregulated. Therefore in the title of the table the word “deregulated” should be changed with “dysregulated”. Moreover, in the table I also suggest replacing the words “upregulated” and “downregulated” with arrow up and arrow down symbols.

Response: Thank you for your insightful recommendation. It has been addressed.

Reviewer 2 Report

Comments and Suggestions for Authors

In this review the authors focused their attention on the complex interplay between epigenetic alterations and OSCC cells. They analyzed the impact of epigenetic modifications on OSCC, including its initiation, progression, and metastasis and their potential role in the treatment and diagnosis of OSCC. 

The review includes sufficient and updated background information. The work is interesting and well written.

I noticed only one inaccuracy: Paragraph 2 "Microenvironment of OSCC"

line 25 pag 6

the authors write: "the extracellular matrix (ECM) is a complex network of macromolecules secreted by tumor cells into the surrounding tissue"

I think the sentence is not completely correct, because the extracellular matrix is not only a complex network of macromolecules secreted by tumor cells, but extracellular matrix is a three-dimensional network of macromolecules, including macromolecules secreted by tumor cells. 

I suggest the authors to better clarify this point. 

Author Response

(Reviewer 2

In this review the authors focused their attention on the complex interplay between epigenetic alterations and OSCC cells. They analyzed the impact of epigenetic modifications on OSCC, including its initiation, progression, and metastasis and their potential role in the treatment and diagnosis of OSCC. 

The review includes sufficient and updated background information. The work is interesting and well written.

I noticed only one inaccuracy: Paragraph 2 "Microenvironment of OSCC"

line 25 pag 6

the authors write: "the extracellular matrix (ECM) is a complex network of macromolecules secreted by tumor cells into the surrounding tissue"

I think the sentence is not completely correct, because the extracellular matrix is not only a complex network of macromolecules secreted by tumor cells, but extracellular matrix is a three-dimensional network of macromolecules, including macromolecules secreted by tumor cells. 

I suggest the authors to better clarify this point. 

Response: Thank you for your insightful recommendation. It has been addressed.

Reviewer 3 Report

Comments and Suggestions for Authors

I have several concerns with the manuscript titled “Epigenetic Regulation in Oral Squamous Cell Carcinoma Microenvironment: A Comprehensive Review” by Mesgari et al.  In its present form, I cannot recommend it for publication as the manuscript is unfocused and, a quick search of the literature finds that reviews on this topic have been published. For example, the authors cite Hena et al. (2017) and a glance at this review reveals that this review has a very similar focus and approach to the topic, but written more concisely.  I think an extensive reworking of the manuscript is needed to be concise, focused, and cohesive.  

1.  There is a great deal of repetition in this manuscript. In many places the authors redefine over and over what is meant by methylation (hypo or hyper). For example, line 272 defines hypomethylation, which is repeated on line 285, again repeated on lines 361-362, etc. It’s as if each section was written by a different person and pieced together into a manuscript.  I found a lack of cohesiveness coupled with the repetition, which makes for a difficult manuscript to read and digest.

2.  The abstract bounces between epigenetics, immunotherapies that can modulate the TME, and development of drug resistance although the title of the manuscript is on epigenetic regulation.  

3.  The introduction is unorganized and is really more of a summary – the first paragraph defines OSCC, the second paragraph is on symptoms, the third paragraph is on diagnosis and treatment, then the forth paragraph goes back to defining OSCC and risk factors, which should have been in paragraph 1.  The fifth paragraph is a sentence – with reference 19 being a similar review with a near identical focus/approach to the topic (but written more concisely) and reference 20 (citation is incomplete) on epigenetics and nothing on potential treatment of OSCC.  This gives me little confidence that the authors are citing sources correctly.  The authors cite work on methylation status of various genes and are not correctly citing the corresponding primary literature.  The authors end the introduction (lines 127-129) with “all aspects of epigenetics in OSCC are thoroughly reviewed and discussed…” yet they spend over half of the manuscript focused on methylation with very little devoted to other epigenetic mechanisms. 

4.  The section on TME (section 2) is not extremely relevant to epigenetic regulation of OSCC given the level of detail they go into in this section. The authors spend time discussing the TME, angiogenesis, tumor development, and antitumor immune responses which detracts from the focus of the paper, especially given the length of the review. This could be handled in the introduction in a small paragraph or two if needed.

5.  The main body of the manuscript (section 3) spends 24 pages discussing which genes are hypomethylated, hypermethylated, or have aberrant methylation and the effect of that methylation state on expression and/or tumor development. This has been done by others, as mentioned above (reference 19) much more succinctly.  Too much detail was given on the function of each of these genes; ach gene discussed spends time focusing on its normal function and what happens when either hypo- or hypermethylated. Some are done succinctly while other genes go into excessive detail.  Here are just a few examples, but by no means is exhaustive:

a.       The section on cyclins only mentions hypomethylation of cyclin D1. There are no epigenetic mechanisms mentioned for other cyclins, other than that these genes are overexpressed in OSCC, which may or may not be due to epigenetics.  

b.       The section on AIM2 spends more time on the cellular implications of AIM2 overexpression versus its epigenetic regulation. The authors postulate that hypomethylation may lead to increased AIM2 expression, and that may be associated with cancer.  

c.       The section on LINE-1 is similar in organization to AIM2. Two pages are devoted to its function when the overall conclusion (lines 454-455) is that LINE-1 hypomethylation leading to cancer development is still unknown.   

d.       The paragraphs (lines 454-489) bring in retrotransposons with LINE-1 (or L1) that seems out of context with the rest of the manuscript. The authors state these retrotransposons lead to genomic instability.  What is the connection between retrotransposons and epigenetic regulation of OSCC?   Just because the promoter region is hypomethylated? 

e.       The authors state twice (line 502 and 515) that the relationship between PI3 kinase and hypomethylation in OSCC is unknown, yet the authors not only discuss this gene in cancer development yet discuss inhibitors of the pathway.  I don’t see how this is relevant to a review on epigenetic regulation.

f.        The authors point out on lines 724-727 the need for IHC standardization. Why?  Because the CDH1 gene shows variation in methylation?  The authors then discuss N-cadherin as a marker for OSCC progression but no mention of any epigenetic regulation.

g.       The section on PTEN includes a large paragraph discussing E-cadherin (lines 772-788).

h.       There is so much known about epigenetic regulation of p53, yet this review is limited to one paragraph (lines 813-819) on HPV. 

6.  Section 4 on therapeutics is interesting but didn’t include a very comprehensive review of these. The section on HDAC inhibitors is incomplete and the authors point the reader to reference 445 in order “To gain and review further information about this topic” which was published in 2010.  There is so much more information out there, and more recent references to cite. Either make this section “a comprehensive review” or consider leaving out.  Same comment for reference 431 (2015) about a review of natural compounds.

7.  The authors should ensure citations are appropriate in the text. There are many places where the citation referenced is either not appropriate or incorrect. Authors should routinely cite the primary literature and not cite review articles unless indicated as such.

Comments on the Quality of English Language

While the English itself is good, there is a lot of repetitive sentences throughout the manuscript. 

Author Response

I have several concerns with the manuscript titled “Epigenetic Regulation in Oral Squamous Cell Carcinoma Microenvironment: A Comprehensive Review” by Mesgari et al.  In its present form, I cannot recommend it for publication as the manuscript is unfocused and, a quick search of the literature finds that reviews on this topic have been published. For example, the authors cite Hena et al. (2017) and a glance at this review reveals that this review has a very similar focus and approach to the topic, but written more concisely.  I think an extensive reworking of the manuscript is needed to be concise, focused, and cohesive.  

  1. There is a great deal of repetition in this manuscript. In many places the authors redefine over and over what is meant by methylation (hypo or hyper). For example, line 272 defines hypomethylation, which is repeated on line 285, again repeated on lines 361-362, etc. It’s as if each section was written by a different person and pieced together into a manuscript.  I found a lack of cohesiveness coupled with the repetition, which makes for a difficult manuscript to read and digest.

Response: Thank you for your insightful comment. It has been addressed. All repetitive phrases and words have been deleted.  

  1. The abstract bounces between epigenetics, immunotherapies that can modulate the TME, and development of drug resistance although the title of the manuscript is on epigenetic regulation.  

Response: It has been addressed. We have modified the Abstract section.

  1. The introduction is unorganized and is really more of a summary – the first paragraph defines OSCC, the second paragraph is on symptoms, the third paragraph is on diagnosis and treatment, then the forth paragraph goes back to defining OSCC and risk factors, which should have been in paragraph 1.  The fifth paragraph is a sentence – with reference 19 being a similar review with a near identical focus/approach to the topic (but written more concisely) and reference 20 (citation is incomplete) on epigenetics and nothing on potential treatment of OSCC.  This gives me little confidence that the authors are citing sources correctly.  The authors cite work on methylation status of various genes and are not correctly citing the corresponding primary literature.  The authors end the introduction (lines 127-129) with “all aspects of epigenetics in OSCC are thoroughly reviewed and discussed…” yet they spend over half of the manuscript focused on methylation with very little devoted to other epigenetic mechanisms. 

Response: Thank you for your insightful comment. These suggestions have been addressed as possible as we could: We changed the end of Introduction section and rechecked citations. Because the topic is extensive, we made truly efforts to concise the information. However, we aimed to address as possible as it does to give a short background about the OSCC and then shift to epigenetics roles. That is why, we found the Introduction section and TME (section 2) best sections to shortly review OSCC. The aim of giving this information was to briefly review OSCC before explaining epigenetics in OSCC (the main subject). Reviewer 1# suggested that some more information about epidemiology of OSCC should be added to introduction section, so we addressed it. And also, Reviewer 2# suggested to explain more about extracellular matrix (ECM) in section 2, so we addressed it. Once again, thank you for your comment.

  1. The section on TME (section 2) is not extremely relevant to epigenetic regulation of OSCC given the level of detail they go into in this section. The authors spend time discussing the TME, angiogenesis, tumor development, and antitumor immune responses which detracts from the focus of the paper, especially given the length of the review. This could be handled in the introduction in a small paragraph or two if needed.

Response: Thank you for your insightful comment. The purpose of this section is to introduce the topic of epigenetics in the context of the TME. This will serve as a foundation for further discussion on the role of epigenetics in TME and its potential implications for cancer development and progression. The TME can directly influence epigenetic modifications in cancer cells and vice versa. For example, immune cells in the TME can release cytokines and other signaling molecules that can modify the epigenetic landscape of cancer cells, affecting their behavior and response to therapy. Epigenetic changes are not limited to cancer cells themselves; they can also occur in cells within the TME. For instance, alterations in the DNA methylation patterns of immune cells or fibroblasts in the TME can impact their function and contribute to tumor immune evasion or the creation of a supportive environment for tumor growth. Understanding these epigenetic modifications in TME cells can provide insights into the interplay between cancer cells and their microenvironment. So this section was thought to help better understand information in the ensuing sections.  

  1. The main body of the manuscript (section 3) spends 24 pages discussing which genes are hypomethylated, hypermethylated, or have aberrant methylation and the effect of that methylation state on expression and/or tumor development. This has been done by others, as mentioned above (reference 19) much more succinctly.  Too much detail was given on the function of each of these genes; ach gene discussed spends time focusing on its normal function and what happens when either hypo- or hypermethylated. Some are done succinctly while other genes go into excessive detail.  Here are just a few examples, but by no means is exhaustive:
  2. The section on cyclins only mentions hypomethylation of cyclin D1. There are no epigenetic mechanisms mentioned for other cyclins, other than that these genes are overexpressed in OSCC, which may or may not be due to epigenetics.  

Response: Thank you for your recommendation. We have mentioned “……within the D-type cyclin family, which includes D1, D2, and D3 isoforms, only cyclin D1 is expressed in cases of oral cancer [97]……”. Then, we have mentioned the impacts of cyclins E, A, and B, those which have been found in OSCC in the following paragraphs.

  1. The section on AIM2 spends more time on the cellular implications of AIM2 overexpression versus its epigenetic regulation. The authors postulate that hypomethylation may lead to increased AIM2 expression, and that may be associated with cancer.  

Response: The current evidence, to the best of our knowledge, has been discussed in this section. We tried to addressed findings related AIM2’s role in OSCC based on epigenetics.

  1. The section on LINE-1 is similar in organization to AIM2. Two pages are devoted to its function when the overall conclusion (lines 454-455) is that LINE-1 hypomethylation leading to cancer development is still unknown.   

Response: The current evidence, to the best of our knowledge, has been discussed in this section. We tried to addressed findings related LINE-1’s role in OSCC based on epigenetics. The precise mechanisms by which LINE-1 hypomethylation influences cancer development are still under investigation, so in this section, it was tried to review current findings briefly.

  1. The paragraphs (lines 454-489) bring in retrotransposons with LINE-1 (or L1) that seems out of context with the rest of the manuscript. The authors state these retrotransposons lead to genomic instability.  What is the connection between retrotransposons and epigenetic regulation of OSCC?   Just because the promoter region is hypomethylated? 

Response: Thank you for your comment. As to our best knowledge and search, several studies like “https://doi.org/10.1186/s12885-021-08174-z, https://doi.org/10.1016/j.cancergen.2020.01.050, and some cited in this sentence” have been found to mention the connection between retrotransposons and epigenetic regulation of OSCC.

  1. The authors state twice (line 502 and 515) that the relationship between PI3 kinase and hypomethylation in OSCC is unknown, yet the authors not only discuss this gene in cancer development yet discuss inhibitors of the pathway.  I don’t see how this is relevant to a review on epigenetic regulation.

Response: Thank you for your comment. The sentence that was mentioned is “While the exact relationship between PI3 kinase and hypomethylation in OSCC is currently unclear, there is evidence to suggest that dysregulation of the PI3K pathway can affect other epigenetic modifications and gene expression patterns.” The authors have mentioned that “exact” mechanism is unclear, but some mechanisms have been found. Thus, some of such mechanisms have been illustrated in this section.  

  1. The authors point out on lines 724-727 the need for IHC standardization. Why?  Because the CDH1 gene shows variation in methylation?  The authors then discuss N-cadherin as a marker for OSCC progression but no mention of any epigenetic regulation.

Response: Thank you for your comment. The reported incidence of CDH1 epigenetic-related modifications in various studies can vary significantly, with values ranging from 7% to 46%. This indicates that more research is needed to better understand and establish consistent findings regarding the epigenetic changes in CDH1.

  1. The section on PTEN includes a large paragraph discussing E-cadherin (lines 772-788).

Response: Thank you for your comment. It has been addressed, and this section has been summarized. 

  1. There is so much known about epigenetic regulation of p53, yet this review is limited to one paragraph (lines 813-819) on HPV. 

Response: Thank you for your comment. Since the aim was to focus on epigenetics applications, it was tried to briefly mention HPV. We think that it is better to deeply focus on HPV in another comprehensive review.

  1. Section 4 on therapeutics is interesting but didn’t include a very comprehensive review of these. The section on HDAC inhibitors is incomplete and the authors point the reader to reference 445 in order “To gain and review further information about this topic” which was published in 2010.  There is so much more information out there, and more recent references to cite. Either make this section “a comprehensive review” or consider leaving out.  Same comment for reference 431 (2015) about a review of natural compounds.

Response: Thank you for your insightful recommendation. It has been addressed and new studies have been added, but it was under consideration to discuss briefly.

  1. The authors should ensure citations are appropriate in the text. There are many places where the citation referenced is either not appropriate or incorrect. Authors should routinely cite the primary literature and not cite review articles unless indicated as such.

Response: Thank you for your insightful recommendation. It has been addressed and all references rechecked and modified.

Comments on the Quality of English Language

While the English itself is good, there is a lot of repetitive sentences throughout the manuscript. 

Round 2

Reviewer 1 Report

Comments and Suggestions for Authors

The authors improved the manuscript addressing all the concerns raised by the reviewer, therefore the article can be accepted for pubblication.

Comments on the Quality of English Language

Moderate editing of English language required